



# Long-term nutrient fertilization and the carbon balance of permanent grassland: any evidence for sustainable intensification?

Dario A. Fornara[1], Elizabeth-Anne Wasson[1], Peter Christie[2], Catherine J. Watson[1]

[1]Agri-Food & Biosciences Institute (AFBI), Newforge Lane, BT9 5PX, Belfast, UK

[2]Institute of Soil Science, Chinese Academy of Sciences, Nanjing 210008, China

*Correspondence to:* Dario Fornara (dario.fornara@afbini.gov.uk)

**Abstract**

Sustainable grassland intensification aims to increase plant yields while maintaining soils' ability to act as sinks rather than sources of atmospheric $CO_2$. High biomass yields, however, from managed grasslands can be only maintained through long-term nutrient fertilization, which can significantly affect soil carbon (C) storage and cycling. Key questions remain about (1) how long-term inorganic *vs.* organic fertilization influences soil C stocks, and (2) how soil C gains (or losses) contribute to the long-term C balance of managed grasslands. Using 43 years of data from a permanent grassland experiment we show that soils not only act as significant C sinks but have not yet reached C saturation. Even unfertilized-control soils showed C sequestration rates of 0.35 Mg C ha$^{-1}$ yr$^{-1}$ (i.e. 35 g C m$^{-2}$ yr$^{-1}$; 0-15 cm depth) between 1970 and 2013. High application rates of liquid manure (i.e. cattle slurry) further increased soil C sequestration to 0.86 Mg C ha$^{-1}$ yr$^{-1}$ (i.e. 86 g C m$^{-2}$ yr$^{-1}$) and a key cause of this C accrual was greater C inputs from cattle slurry. However, average coefficients of 'Slurry-C retention' suggest that 85% of C added yearly through liquid manure is lost possibly via $CO_2$ fluxes and organic C leaching from soils. Inorganically fertilized soils (i.e. NPK) had the lowest 'C-gain-efficiency' (i.e. unit of C gained per unit of N added) and lowest C sequestration (similar to control soils). Soils receiving cattle slurry showed higher C-gain and N-retention efficiencies compared to soils receiving NPK or pig slurry. We estimate that net rates of $CO_2$-sequestration in the soil top 15 cm can offset 9-to-25% of GHG emissions from intensive management. However, because of multiple GHG sources associated with livestock farming, the net C balance of these grasslands remains positive (9-to-12 Mg $CO_{2\text{-equivalent}}$ ha$^{-1}$ yr$^{-1}$), thus contributing to climate change. Further C-gain efficiencies (e.g. reduced enteric fermentation and use of feed concentrates, better nutrient-management) are required to make grassland intensification more sustainable.

**Keywords**: Carbon sequestration, GHG emissions, livestock, Global Warming Potential, sustainability

## 1. Introduction

Grassland soils are extremely important to human society not only because they provide a range of goods and services that support the livelihoods of ~ 1 billion people worldwide (Suttie et al., 2005) but also because the good functioning of soil ecosystems is fundamental to the long-term sustainability of agricultural economies and the livestock sector (Tilman et al., 2002; Alexandratos and Bruinsma, 2012). From a global change perspective managed grassland soils have received much attention over the last two decades for two important reasons: (1) the livestock sector is responsible for approximately 14.5% of all anthropogenic GHG emissions worldwide



(Gerber et al., 2013), and (2) permanent grassland soils have the ability to sequester carbon (C) thereby partly offsetting C emissions (Fornara et al., 2011, 2013). Evidence from multiple European grassland sites show that soil C sequestration rates may reach 0.05 Mg C ha$^{-1}$ yr$^{-1}$ according to inventories of soil organic C stocks and 0.77 Mg C ha$^{-1}$ yr$^{-1}$ for mineral soils according to C flux balance measurements (Soussana et al., 2010). Similar observations using net $CO_2$ exchange measurements from flux towers provide estimates of soil C sequestration rates in grasslands equal to 0.57 Mg C ha$^{-1}$ yr$^{-1}$ (Schulze et al., 2009). A meta-analysis of numerous global grassland studies has reported potential C sequestration rates of 0.22 Mg C ha$^{-1}$ yr$^{-1}$ (Smith et al., 2008). Evidence from the process-based biogeochemical model ORCHIDEE-GM (Chang et al., 2015) shows C sequestration rates (biomass + soils) of 0.15 Mg C ha$^{-1}$ yr$^{-1}$ in European grasslands over a period of 50 years (1961–2010). Data from long-term soil surveys and repeated sampling studies (Bellamy et al., 2005; CLIMSOIL, 2008; Smith, 2014) show either C losses or gains from topsoils across different grassland types.

Altogether these studies suggest that grassland soils under human management can potentially act as significant C sinks; it is not clear, however how soil C sequestration will ultimately contribute to the net C balance of intensively managed grasslands especially when we account for large GHG emissions from the livestock sector (i.e. emissions from enteric fermentation of ruminant livestock, manure management etc.). Thus the main aims of this study are to (1) estimate rates of soil C sequestration under long-term (43 years) additions of either inorganic synthetic fertilizer (NPK) or organic nutrients (i.e. cattle and pig liquid manure) to permanent grassland, and (2) assess to what extent soil's ability to accumulate C will actually offset overall C losses (in $CO_{2-equivalents}$) associated with intense grassland management.

Across Europe grassland management intensification involves increased nutrient fertilization, greater livestock density and higher mowing frequency (Soussana and Lemaire, 2014; Manning et al., 2015). There is increasing evidence over the last 15 years that GHG emissions from synthetic fertilizer applications have been growing faster than any other agriculture-related sources and that synthetic fertilizers may become the second largest emission source after enteric fermentation over the next decade (FAO, 2014). There is also evidence that agriculture's contribution to GHG emissions may be now larger than emissions from the land use sector mainly because the use of fertilizers and animal manures contribute substantially to climate change through their energy intensive production and inefficient use (FAO, 2014). Finally, it has been shown that both inorganic (Fornara et al., 2013; Cenini et al., 2015) and organic nutrient fertilization (De Bruijn et al., 2012; Maillard and Angers, 2014) could significantly influence soil C sequestration.

Based on this evidence the question remains how to make grassland management intensification more sustainable when (1) long-term nutrient fertilization remains crucial to maintain soil fertility and plant yields (Smith et al., 2015), (2) permanent grasslands are mown more frequently and most aboveground productivity is removed thus not contributing to C inputs to soils, and the demand for milk and meat production is increasing.

Recent research findings (Hristov et al., 2013; FAO, 2014) show that increases in commodity outputs per animal or per unit land area (i.e. kg cattle meat per hectare) in intensive livestock production systems can be larger than the corresponding increases in GHG emissions per animal or per unit land area. These findings do not suggest that intensification of production is a mitigation strategy *per se* but rather advocate more research on what C efficiency gains could help in improving the C-footprint of different production systems. In this study we mainly focus on potential soil C gains in a 43-year-long grassland experiment asking (1) how changes in soil C stocks might be affected by long-term inorganic and organic nutrient fertilization, and (2) to what extent soil





C gains (or losses) might contribute to the C-footprint (i.e. net long-term C balance) of this human-managed system.

## 2. Materials and Methods

### 2.1. Site description

The long-term (43-year-old) grassland experimental site is located at Hillsborough, County Down, Northern Ireland (Irish Grid Reference J 244577). The Long-Term Slurry experiment (LTS) was established in 1970 on a pre-existing sward of perennial ryegrass at an elevation of about 120 m above sea level. Mean annual precipitation and temperatures in this area between 2000 and 2015 are ~900 mm and 9 ℃ respectively (www.metoffice.gov.uk). The soil is a clay loam (42% sand, 24% silt and 34% clay) overlying Silurian shale and greywacke (Christie, 1987). LTS was designed to measure the effects of frequent applications of organic animal slurries and synthetic fertilizer on grass yields both in terms of grass quality and quantity and on physical and chemical properties of the soil. LTS receives eight nutrient treatments (see Fig. 1 in Supporting Information) as follows: unfertilized control, fertilized control (i.e. NPK at rates of 200 kg N, 32 kg P, 160 kg K ha$^{-1}$ year$^{-1}$), pig slurry at 50, 100 or 200 m$^3$ ha$^{-1}$ year$^{-1}$, and cattle slurry at the same three increasing rates. Hereafter, we refer to low (L), medium (M) and High (H) rates of pig slurry applications (i.e. 50, 100 or 200 m$^3$ ha$^{-1}$ year$^{-1}$) as 'Pig (L), (M) and (H), respectively. Similarly, we refer to Cattle (L), (M) and (H). There are six replicates, giving 48 plots in a randomized block design (Fig. 1 in Supplementary Material). The NPK synthetic fertilizer and slurries have been applied every year (since 1970) in three equal dressings, first in the spring and then immediately after the first two cuts. Grass is cut three times every year at the silage stage.

Previous botanical surveys carried out at different time intervals (in 1981-82 and 2006) show how plant species composition has remained remarkably stable in the slurry treated swards over the 24-year-gap between the two surveys (Christie, 1987; Liu et al., 2010). Common C3-grass species namely *Lolium perenne*, *Agrostis stolonifera* and *Poa spp.* account for at least 70% of the aboveground plant biomass in plots either receiving inorganic and organic nutrient fertilization. Three C3 grasses also account for >70% of aboveground biomass in the control plots, namely *Lolium perenne*, *Holcus lanatus* and *Agrostis tenuis*. Legume species mainly represented by *Trifolium repens* account for ~5% of aboveground biomass in the control plots and ~0.05% or less in the nutrient fertilized plots (Liu et al., 2010).

### 2.2. Soil, root and animal slurry sampling and analyses

Soil samples were collected in 1972, 1975, 1986, 1988, 1991, 1996, 2001, 2007, 2012 and 2013 at three depth-intervals (0-5, 5-10, 10-15 cm) and at 8 random locations within each experimental plot (29.75 m$^2$ each) in February each year using a 3 cm diameter soil corer. Soils were sieved through 2 mm mesh-size, air dried, ground to < 150 μm and analyzed for total C% and N% using a LECO Trumac CN Analyser (St Joseph, MI, USA). To determine organic C concentration, bulk soil subsamples were burned for 16 h at 550 °C in a muffle furnace. In 2015, eight soil cores (3 cm diameter) were collected in each plot to 20 cm depth to estimate total root mass. These soil samples were washed gently with water over a fine mesh screen until roots were free of soil. Roots were then dried at 50 °C to constant mass and weighed. Two larger soil cores were collected in each plot at two different soil depths (0-10 and 10-20 cm) to measure soil bulk density as the ratio between air-dried (stone-free) soil and soil volume. Soil bulk density was measured in 1970 on a grassland field beside the LTS



experimental site and re-measured in 2013 in each LTS experimental plot. Triplicate samples of fresh slurry were first dried on a steaming water bath to drive off volatile moisture before being placed in an oven at 105 °C for 16 hours. The oven dry slurry was then cooled in a desiccator and then milled using a Culatti hammer mill (Glen Creston Ltd) in preparation for total C% and N% measurements using the LECO analyser.

### 2.3. Plant biomass productivity

Grass biomass was harvested 3 times every year (usually in May, July and September). Each experimental plot has a harvestable area of 18.75 $m^2$ within which all grass biomass is cut at a height of 5 cm with a plot harvester. Total fresh plant biomass is weighed in the field at each harvest and a sub-sample of 300 g fresh weight is then oven dried at 80°C to estimate dry matter production per hectare per year under each nutrient treatment.

### 2.4. Long-term C balance of permanent grassland

To estimate the long-term net C balance of our permanent grassland we calculated the Global Warming Potential (GWP) associated with each experimental nutrient treatment (Robertson et al., 2000; Fornara et al., 2011). GWP gives an estimate of the cumulative radiative forcing of individual greenhouse gases relative to a reference gas (i.e. $CO_2$), over a specific time span, here 100 years. The GWP of each nutrient management practice in these grassland soils was estimated based on the contribution of several GHGs ($CH_4$, $CO_2$ and $N_2O$-N) to emissions. Our aim here was to estimate GWP at the farm-level under common management practices ('business as usual') for dairy or beef farming systems under Western-European environmental and economic conditions. Gains and losses of $CO_{2-eq}$ were calculated from (see Supporting Information for details): (1) field application of calcareous lime ($CaCO_3$), (2) production and transportation of $CaCO_3$, (3) enteric fermentation of ruminant livestock, (4) manure management ($CH_4$), (5) manure management ($N_2O$-N), (6) managed soils ($CH_4$), (7) managed soils ($N_2O$-N), (8) milk yields, (9) beef yields, (10) feed concentrate production and transportation, (11) fertiliser production, and (12) machinery use. We also estimated microbial oxidation of $CH_4$ in $CO_2$-eq and net C sequestration rates in soils under each nutrient treatment between 1970 and 2013.

### 2.5. Data Analysis

To test for the potential effects of different nutrient treatments on soil C and N sequestration we initially performed ANOVA analyses on log-transformed soil data collected from the 2 x 3 factorial design, which includes slurry type × rate of slurry application. We also compared means between NPK and control treatments and between NPK (and control) and overall mean of the cattle and pig slurry treatments. The least significant difference applicable to this comparison was used to make a judgment about the closeness of NPK (and control) to either the mean for slurry type or slurry rate. Means of each slurry type at a given rate were only considered if the type × rate interaction was significant. We performed repeated measures analysis of variance (MANOVA) using log-transformed data from 1972, 1975, 1986, 1988, 1991, 1996, 2001, 2007, 2012 and 2013. C (and N) stocks between 0-10 cm soil depth were calculated by using soil bulk density values for the 0-10 cm layer and soil C% (or N%) between the 0-5 and 5-10 cm layers. C (and N) stocks between 10-15 cm soil depth were calculated by using soil C or N content (in the same 10-15 cm layer) and soil bulk density of the 10-20 soil layer. We assumed that changes in soil bulk density between 0-10 cm depth and between 10-20 cm soil depth were negligible as we found evidence of this from mean bulk density values calculated across the 0-20 cm soil





layer (see Results). We calculated the difference in total soil C (and total soil N) content between each fertilized plot and the average of the unfertilized controls, and then divided this value by the N added (from each of the animal slurry applications and from the NPK treatment) to that plot over 43 years. We thus estimated (1) the units of soil C gained under each nutrient treatment per unit of N added (i.e. "Net C gain per unit of N added" expressed as: g C g$^{-1}$ N added), and (2) the units of soil N retained per unit of N added either through animal manure or through NPK additions (i.e. "Net N retention per unit of N added" expressed as: g N g$^{-1}$ N added). We also calculated the 'Slurry-C retention coefficient' (*sensu* Maillard and Angers, 2014), which is the average proportion of slurry-C, which has been annually 'retained' in soils after 43 years of slurry applications. Data were analyzed using JMP version 9.0.0 (SAS Institute, Cary, North Carolina, USA).

## 3. Results

### 3.1. Long-term changes in soil C and N stocks

We found that soil bulk density averaged between 0-10 cm depth was very similar across nutrient treatments with values ranging between 1.027 and 1.039 g cm$^{-3}$ thus <1% variation across our soils after 43 years of different nutrient applications. Similarly, soil bulk density measured between 10-20 cm ranged between 1.025 and 1.031 g cm$^{-3}$ (<1% variation across nutrient treatments). Our reference value for soil bulk density in 1970 was equal to 1.1, a value, which was used to calculate soil C stocks for 1970 and 1975. We found that soil C stocks significantly increased under all nutrient management treatments including the control (Fig. 1). Soil C stocks increased linearly within each experimental treatment. Results from repeated measures MANOVA showed a positive significant effects of time ($F_{9,32}$ = 141; P<0.0001) and time × treatment interaction ($F_{63,186}$ = 3.67; P<0.0001). Nutrient treatment had a highly significant effect on soil C accumulation ($F_{7,40}$ = 13.8; P<0.0001). Similarly we found positive significant effects of time (P<0.0001) and time × treatment interaction (P<0.0001) on changes in soil N stocks across our experimental plots (see Fig. 2 in Supplementary Material).

The highest rate of soil C sequestration was observed under the high cattle slurry applications ('Cattle (H)' at 200 m$^3$ of slurry ha$^{-1}$ year$^{-1}$; Fig. 2a) and was equivalent to 0.86 ± 0.05 Mg C ha$^{-1}$ yr$^{-1}$. Relative high rates of soil C sequestration were also observed under low 'Cattle (L)' (i.e. 50 m$^3$ slurry ha$^{-1}$ year$^{-1}$) and medium 'Cattle (M)' (i.e. 100 m$^3$ slurry ha$^{-1}$ year$^{-1}$) slurry treatments which sequestered 0.42 ± 0.02 and 0.55 ± 0.04 Mg C ha$^{-1}$ yr$^{-1}$, respectively. Applications of pig slurries led to lower rates of soil C accumulation compared to cattle slurries (Fig. 2a), being 0.37 ± 0.04, 0.28 ± 0.03 and 0.31 ± 0.03 Mg C ha$^{-1}$ yr$^{-1}$ for low, medium and high (i.e. Pig (L), (M) and (H)) application rates, respectively. Although not significant, control plots accumulated more C in the soil than NPK-fertilized plots (0.35 ± 0.03 and 0.32 ± 0.03 Mg C ha$^{-1}$ yr$^{-1}$).

The highest rate of soil N sequestration was also observed under the Cattle (H) treatment (Fig. 3 in Supplementary Material) and was equivalent to 0.08 ± 0.003 Mg N ha$^{-1}$ yr$^{-1}$. Medium cattle slurry applications (Cattle (M)) sequestered 0.05 ± 0.003 Mg N ha$^{-1}$ yr$^{-1}$, whereas Cattle (L) sequestered as much as Pig (H) which was 0.03 ± 0.003 Mg N ha$^{-1}$ yr$^{-1}$. Although not significant the NPK treatment sequestered more N (i.e. 0.019 ± 0.003 Mg N ha$^{-1}$ yr$^{-1}$) than the control treatment (i.e. 0.011 ± 0.003 Mg N ha$^{-1}$ yr$^{-1}$; Fig. 3 in Supplementary Material).

In multiple regression analyses where we included different independent variables (e.g. root mass, root C:N ratio, grass biomass yields, C inputs from slurry etc.) we found that only C inputs from animal slurry significantly positively affected (P < 0.0001) net soil C sequestration rates over 43 years (see Fig. 2b). We found





that the highest level of cattle slurry application significantly increased soil C (%) at all three soil depths when compared to the other nutrient treatments and the control (Fig. 3). Under the highest cattle slurry applications soil C content, averaged for two recent years (2012 and 2013), was $8.7 \pm 0.2\%$ (0-5 cm depth; $P < 0.0001$), $5.83 \pm 0.3\%$ (5-10 cm depth; $P < 0.0001$) and $3.94 \pm 0.1\%$ (10-15 cm depth; $P < 0.01$). No significant differences were found in soil C% (and N%; Supplementary Fig. 4) content between the control, NPK and the three pig slurry application treatments.

We did not find any significant difference in soil C stocks (15-30 cm depth) between nutrient treatments in 1998 ($F_{6,34} = 1.61$, $P = 0.21$) and in 2013 ($F_{7,42} = 1.03$, $P = 0.42$; Supplementary Fig. 5a, b).

Finally, we found that soil C:N ratios were significantly higher in the control plots (i.e. C:N = 12.75; $P < 0.0001$; Tukey-Kramer test <0.05) compared to the other treatments and that Cattle (M) and (H) and Pig (H) had the lowest soil C:N ratios being 10.9, 10.7 and 10.6, respectively.

### 3.2. Net soil C retention per unit of C and N added

The rate of C addition to soils was calculated for each of the different animal slurry applications (Table 1). We found that cattle slurry applications of 50, 100 and 200 $m^3$ of slurry $ha^{-1}$ $year^{-1}$ contributed to add 0.92, 1.84 and 3.67 Mg C $ha^{-1}$ $yr^{-1}$ to the soil, respectively while pig slurry applications contributed 0.27, 0.55 and 1.1 Mg C $ha^{-1}$ $yr^{-1}$, respectively. Based on these yearly C inputs (for 43 years) we found that the highest 'Slurry-C retention coefficients' were associated with Cattle (H) ($0.15 \pm 0.04$ g C $g^{-1}$ C added) and Pig (L) ($0.16 \pm 0.04$ g C $g^{-1}$ C added) treatments (Table 1). This meant that approximately 15-16% of the C applied every year through animal manure remained in the soil. The 'Slurry-C retention coefficient' is an index of the ability of the grassland soils to retain any C added through animal manure. The relative change in soil C stocks (i.e. the ratio of SOC stock in plots receiving slurry to SOC stock in plots receiving NPK) was higher under cattle slurry than pig slurry amendments (see Table 1).

We also calculated N input rates from animal slurry and found that Cattle (L), (M) and (H) treatments contributed 0.16, 0.32 and 0.64 Mg N $ha^{-1}$ $yr^{-1}$ to the soil, respectively, while pig slurry applications contributed 0.13, 0.27 and 0.54 Mg N $ha^{-1}$ $yr^{-1}$, respectively. We then calculated soil C retention rates per unit of N added based on these N inputs from slurries and from NPK (i.e. 0.2 Mg N $ha^{-1}$ $yr^{-1}$). We found that the highest rates of soil C retention per unit of N added were associated with Cattle (M) and (H) ($0.88 \pm 0.11$ and $0.92 \pm 0.11$ g C $g^{-1}$ N added; Fig. 4a). We found lower soil C retention rates associated with other slurry treatments and evidence of even negative rates of soil C retention (i.e. grams of soil C lost per g of N added) under the NPK treatment (Fig. 4a). Similarly we observed higher rates of soil N retention per unit of N added associated with Cattle (M) and (H) ($0.10 \pm 0.02$ and $0.11 \pm 0.03$ g N $g^{-1}$ N added) when compared to NPK and pig slurry applications (Fig. 4b).

### 3.3. Plant aboveground and belowground biomass

We found that total annual plant yields as averaged for five recent years (2009 to 2013) were significantly higher under the highest rates of cattle ($19 \pm 0.3$ Mg dry mass $ha^{-1}$) and pig ($20 \pm 0.4$ Mg dry mass $ha^{-1}$) slurry applications when compared to other treatments (Fig. 5a). Plant yields were significantly lower in the control plots ($2.8 \pm 0.01$ Mg dry mass $ha^{-1}$; Fig. 5a). In contrast, total root mass significantly declined with increasing





slurry application rates being $0.92 \pm 0.2$ Mg dry root mass for Cattle (H) and $1.1 \pm 0.2$ Mg dry root mass for Pig (H). Root mass was highest in the control plots ($3.15 \pm 0.3$ Mg dry root mass ha$^{-1}$).

### 3.4. Soil C sequestration and GWP of permanent grassland

We found that the GWP associated with 'business as usual' dairy or beef farm management was always positive thus contributing to net GHG emissions and to Global Climate Change. GWP estimated for dairy farming varied between 1162 to 1201 g $CO_{2-eq}$ m$^{-2}$ yr$^{-1}$ (i.e. 11.6 to 12 Mg $CO_{2-eq}$ ha$^{-1}$ yr$^{-1}$) under increasing cattle slurry applications and 1246 g $CO_{2-eq}$ m$^{-2}$ yr$^{-1}$ (i.e. 12.5 Mg $CO_{2-eq}$ ha$^{-1}$ yr$^{-1}$) under NPK applications. GWP estimated for beef farming ranged from 913 to 952 g $CO_{2-eq}$ m$^{-2}$ yr$^{-1}$ (i.e. 9.1 to 9.5 Mg $CO_{2-eq}$ ha$^{-1}$ yr$^{-1}$) under increasing

cattle slurry applications and 997 g $CO_{2-eq}$ m$^{-2}$ yr$^{-1}$ (i.e. 9.9 Mg $CO_{2-eq}$ ha$^{-1}$ yr$^{-1}$) under NPK applications. These are net $CO_{2-eq}$ emissions after removing $CO_2$ sequestered in soils under each of the different long-term nutrient application treatments. Based on these values we estimate that soil $CO_2$ sequestration contributes to offset overall GHG emissions between 9% (under NPK additions) and 20.8% (under highest cattle slurry applications) in dairy farms and between 10.5% (NPK additions) and 25% (cattle 'H' treatment) in beef farms (Fig 6).

### 4. Discussion

Overall our results show that permanent grassland soils have not reached C saturation after 43 years of intensive management (i.e. three grass cuts and three nutrient applications every year since 1970). Even soils not receiving any nutrient addition showed C sequestration rates of 0.35 Mg C ha$^{-1}$ yr$^{-1}$ (35 g C m$^{-2}$ yr$^{-1}$) within 0-15

cm soil depth. The application of high rates of cattle slurry significantly contributed to increase soil C sequestration up to 0.86 Mg C ha$^{-1}$ yr$^{-1}$ (86 g C m$^{-2}$ yr$^{-1}$), whereas the application of pig slurry and inorganic NPK fertilizer did not have any significant C sequestration benefit when compared to unfertilized (control) soils. Carbon sequestration rates in the current study are within the range of average soil C accumulation rates estimated across managed grasslands in Europe (Watson et al., 2007; Smith et al., 2008; Schulze et al., 2009;

Soussana et al., 2010; Chang et al., 2015). Thus our evidence is that intensively managed (permanent) grassland soils not only can act as C sinks but do not show any significant C loss under long-term nutrient applications and frequent grass mowing. Our results also suggest that the application of cattle excreta to grassland is a key factor sustaining soil C gains (De Bruijn et al., 2012; Maillard and Angers, 2014; Hunt et al., 2016) relatively to C losses.


### 4.1. Organic vs. inorganic fertilization and soil C sequestration

Our results show that cattle slurry applications are more effective than pig slurry applications in increasing C accumulation in top-soils and this positive effect could have multiple explanations. First, the amount of C added to soils through cattle slurry is higher than in pig slurry (Table 1), and we found a strong positive correlation

between C inputs to soils and net soil C accumulation (Fig. 2b). Our estimate is that cattle slurry-C retention efficiency varies between 14-15%, a value that is very similar to a global manure-C retention coefficient of 12% estimated in a recent meta-analysis study (Maillard and Angers, 2014). This means that 85-86% of all C applied yearly to our soils is lost potentially via increased soil $CO_2$ fluxes and/or organic C leaching from soils. However, 15% C retention efficiency associated with high rates of cattle slurry significantly contributed to soil

C accrual in this long-term grassland experiment.



A second reason why cattle slurry could be more beneficial to soil C storage may be due to its biochemical composition. We did not find any significant difference in total N% (2.9 ± 0.9 vs. 3.1 ± 0.7) or total C% (32.3 ± 2.8 vs. 34.8 ± 3.7) content between pig and cattle slurry, respectively. However, animal ruminants such as cattle are particularly efficient in using C components in their grass feed and excrete high concentrations

of slowly digestible organic matter including lignocelluloses (Van Kessel et al., 2000). It is well established that lignin concentrations are negatively related to the biodegradability of organic material in animal manures (Triolo et al., 2011), and that high lignin-to-N ratios tend to slow organic transformations in soils (Chantigny et al., 2002) and can thus lead to greater organic C accumulation in grassland soils (Parton et al., 1987). Recent evidence, however suggests that the concentration of lignin and other biochemical compounds either in animal

slurries or in different organic amendments are not necessarily good predictors of their long-term residence time in soils (Lashermes et al., 2009).

Thirdly, long-term pig slurry applications were generally associated with poor soil C and N retention per unit of N added (Fig. 4a, b). Here soil C and N retention rates were higher under low rates of pig slurry applications suggesting that increasing application rates of pig slurry may contribute to higher C losses from

soils. Previous studies have shown that pig slurry applications can cause soil C losses either by significantly increasing soil $CO_2$ fluxes (Rochette et al., 2000) or by promoting a 'priming effect' whereby high biodegradable pig slurry applications stimulate the mineralization of both native soil C and fresh root-derived material (Angers et al., 2010). Given the high variability in the concentration and abundance of both recalcitrant (e.g. lignin) and labile (e.g. volatile fatty acids) organic compounds in pig and cattle slurries more studies are

required to address potential linkages between slurry biochemistry and long-term C and N transformations in soils.

In the current study, cattle slurry led to greater C and N soil retention rates per unit of N added, than inorganic synthetic fertilizer. Soils receiving high rates of cattle slurry retained 0.89-0.94 grams of C per gram of N added (Fig. 4a) whereas soils receiving NPK additions did not retain any C per unit of N added. This may

explain why NPK-fertilized soils did not show any significant change in C stocks when compared to the unfertilized control. The results that long-term NPK fertilization does not have any additional benefit for soil C sequestration, is supported by Fornara et al., (2013), who showed that if chronic multi-nutrient additions greatly contributed to increased grass yields, they also significantly decreased total root mass and did not have any effect on soil C stocks compared to unfertilized soils. Our study also suggests that different processes might

drive long-term soil C accumulation, for example, in soils amended with cattle slurry it is clear that C accumulation is strongly increased by 'external' C additions through liquid manure. As opposite in the control plots which do not receive any C addition, soil C accumulation may be partly explained by higher root mass (possibly resulting from positive grass-legume interactions), which contributes to higher C inputs to soils when compared to the reduced root systems of fertilized plots (see Fig. 5b).

An important finding of our study is that long-term effects of nutrient fertilization on soil C stocks were detected between 0-15 cm depth only. Our evidence is that differences in soil C (%) content among nutrient treatments greatly decreased between 10-15 cm soil depth (Fig. 3) and that soil C stocks measured in 1998 and 2013 between 15 and 30 cm depth were not significantly different among nutrient treatments (Supplementary Fig. 5). Interestingly this finding supports the long-term predictions of a recent model

simulation, which shows how C accumulation (in the first 100 years since grassland establishment) mainly



occurs in the top 15 cm of the soil profile (De Bruijn et al., 2012). IPCC guidelines define soil carbon stocks as organic C incorporated into mineral soil horizons to a depth of 30 cm; it might be, however that net rates of atmospheric $CO_2$ sequestration in the top 15 cm only of the soil profile could give reliable estimates of soils' capacity to offset GHG emissions within a time scale of 100 years since grassland establishment (see below).

### 4.2. Soil C sequestration and the long-term balance of permanent grassland

The Global Warming Potential (GWP) of our permanent grassland is an estimate of the net balance between long-term soil C gains (i.e. soil C sequestration) and overall C losses calculated (in this case) at the farm-level under common grassland and livestock management activities (see Supporting Information). Our results show
relatively high GWP values associated with farm management suggesting that these livestock systems contribute to global climate change by emitting between 9 and 12 Mg $CO_{2-eq}$ ha$^{-1}$ yr$^{-1}$. We are not aware of any other study, which has quantified net C balance of permanent grassland under long-term inorganic and organic nutrient fertilization. However, our estimates are comparable with the range of net emissions (i.e. GWPs) of 7.5-to-11.2 Mg $CO_{2-eq}$ ha$^{-1}$ yr$^{-1}$, associated with intensively managed permanent grasslands in the Allgäu region of
Germany (Haas et al., 2001). Moreover, if we consider GWP values in relation to the production of an important agricultural commodity (i.e. milk) we estimate that the C-footprint of milk production in our grassland ranges between 0.96 to 1.03 Mg $CO_{2-eq}$/ Mg milk, which is similar to average values estimated from across many farms under similar environmental conditions (DairyCo, 2014; O'Brien et al., 2014). Our evidence is that long-term $CO_2$ sequestration in grassland soils can offset between ~9% and ~25% of overall GHG emissions estimated at
the 'farm-gate' for dairy and beef farms (Fig. 6). Dairy cattle management at the farm-level is less able to offset GHGs when compared to beef management simply because dairy cattle are associated with higher $CH_4$ emissions from enteric fermentation and manure management. Our estimates of soils' ability to offset overall GHG emissions agree with average values obtained by the carbon-foot print calculator 'BovIs' developed by the Agri-Food Biosciences Institute (see Aubry et al., 2013), validated by comparing GHG emissions from a
hundred farms across Northern Ireland.

### 5. Conclusion

To our knowledge this study is the first to simultaneously show (1) how long-term (i.e. 43 years) applications of inorganic and organic nutrients can influence C sequestration in permanent grassland soils, and (2) to what
extent soil C accrual can offset GHG emissions from livestock-based farming systems. Our findings suggest that permanent grassland soils act as sinks rather than sources of atmospheric $CO_2$ and that there was no evidence of an equilibrium state having been reached after 43 years. Even unfertilized control plots had C sequestration rates of 0.35 Mg C ha-1 yr$^{-1}$. Application of pig slurry and inorganic NPK fertilizer did not lead to any further C sequestration. However, high rates of cattle slurry significantly increased C sequestration rates to 0.86 Mg C ha$^-$
$^1$ yr$^{-1}$. The results showed that soil C sequestration does contribute to reducing the C-footprint of key commodities (i.e. milk). However, to make grassland intensification more sustainable significant C-gains need to be made by reducing emissions from enteric fermentation, manure management and from the production and transport of feed concentrates, which are used to supplement animal diets.






**Acknowledgements**

This research was funded by the Department of Agriculture and Rural Development (DARD) of Northern Ireland (UK) - project number 7001 (41499). We would like to thank Alan Wright, Elizabeth Mulligan, Brian Wallace, David Flynn and Gareth Ridgway for their assistance with field sampling and laboratory analysis.

Scott Laidlaw provided useful comments, which greatly improved the manuscript.

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




**Table 1.** Effects of C application rates of different animal slurries to soils on (1) Slurry C-retention coefficient, and on (2) Relative soil C stocks change. The 'Slurry-C retention coefficient' represents the average proportion of slurry-C, which has been annually 'retained' in soils after 43 years of slurry applications. The 'Relative soil C stocks change' is the ratio between SOC stocks in plots receiving slurry and SOC stocks in plots receiving mineral (i.e. NPK) fertilization (*sensu* Maillard and Angers, 2014).

| Animal Slurry Treatment [(*)] | C application rates (Mg C ha$^{-1}$ yr$^{-1}$) | (1) Slurry C-retention coefficient ($\pm$ s.e.) | (2) Relative soil C stocks change ($\pm$ s.e.) |
|---|---|---|---|
| Pig (L) | 0.27 ± 0.02 | 0.16 ± 0.04 | 1.06 ± 0.007 |
| Pig (M) | 0.55 ± 0.06 | 0.03 ± 0.07 | 1.02 ± 0.02 |
| Pig (H) | 1.11 ± 0.12 | 0.07 ± 0.06 | 1.05 ± 0.04 |
| Cattle (L) | 0.92 ± 0.1 | 0.14 ± 0.06 | 1.09 ± 0.03 |
| Cattle (M) | 1.84 ± 0.2 | 0.14 ± 0.03 | 1.17 ± 0.03 |
| Cattle (H) | 3.67 ± 0.45 | 0.15 ± 0.01 | 1.36 ± 0.02 |

[(*)] Slurry pig (L), (M) and (H) = pig manure at 50, 100 and 200 m$^3$ ha$^{-1}$ yr$^{-1}$, respectively; Slurry cattle (L), (M) and (H) = cattle manure at 50, 100 and 200 m$^3$ ha$^{-1}$ yr$^{-1}$, respectively.

20

25





**Fig. 1.** Changes in soil C stocks from 1970 to 2013 (0-15 cm soil depth) under either organic or inorganic nutrient additions or no-nutrients (i.e. control). Nutrient application rates: NPK = 200 kg N, 32 kg P, 160 kg K ha$^{-1}$ yr$^{-1}$; Pig (L), Pig (M) and Pig (H) = Pig slurry applications at 50 (Low), 100 (Medium) and 200 (High) m$^3$ ha$^{-1}$ yr$^{-1}$, respectively; Cattle (L), Cattle (M), Cattle (H) = Cattle slurry applications at 50 (Low), 100 (Medium) and 200 (High) m$^3$ ha$^{-1}$ yr$^{-1}$ respectively. Best fit linear functions: NPK (y = 0.36x - 647; R² = 0.87); Control (y = 0.41x - 746; R² = 0.83); Pig (L) (y = 0.39x - 723; R² = 0.88); Pig (M) (y = 0.28x − 498; R² = 0.65); Pig (H) (y = 0.31x − 546; R² = 0.67); Cattle (L) (y = 0.43x − 796; R² = 0.91); Cattle (M) (y = 0.65x − 1230; R² = 0.95); Cattle (H) (y = 0.86x − 1636; R² = 0.98).

**Fig. 2.** Relationship between net soil C sequestration rates (Mg C ha$^{-1}$ yr$^{-1}$; 0-15 cm depth) and (a) nutrient application treatments, (b) C inputs from animal slurry (Mg C ha$^{-1}$ yr$^{-1}$). Experimental treatments same as in Fig. 1.

**Fig. 3.** Relationship between mean soil C (%) content at three soil depths (A=0-5 cm; B=5-10 cm; C=10-15 cm) and different nutrient treatments. Abbreviations as in Fig. 1.

**Fig. 4.** Relationships between (a) net soil C gain (i.e. grams of C 'gained' in the soil) and (b) net soil N retention (i.e. grams of N 'retained' in the soil) per gram of N added over 43 years under each nutrient treatment. Abbreviations as in Fig. 1.

**Fig. 5.** Relationships between (a) grass yields (averaged for five recent years 2009-2013) and (b) total root mass (measured in 2014 between 0-20 cm soil depth) and the different nutrient treatments. Abbreviations as in Fig. 1.

**Fig. 6.** Percentage of GHG emissions (in CO$_{2-eq}$) that can be offset by soil C sequestration associated with long-term inorganic and organic nutrient additions. Estimates refer to dairy and beef farming systems and are calculated as net contribution of multiple GHGs (i.e. Global Warming Potential; see Supporting Information). Abbreviations as in Fig. 1.





**Fig. 1**

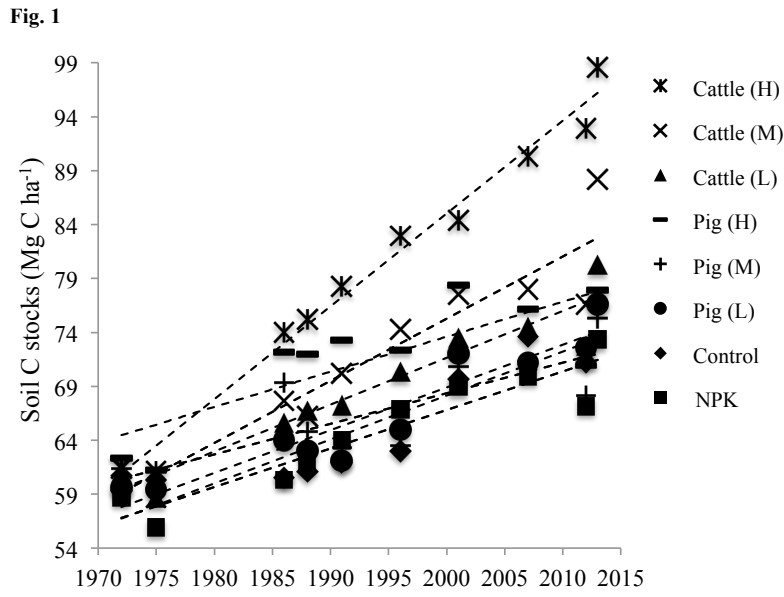

5 **Fig. 2**

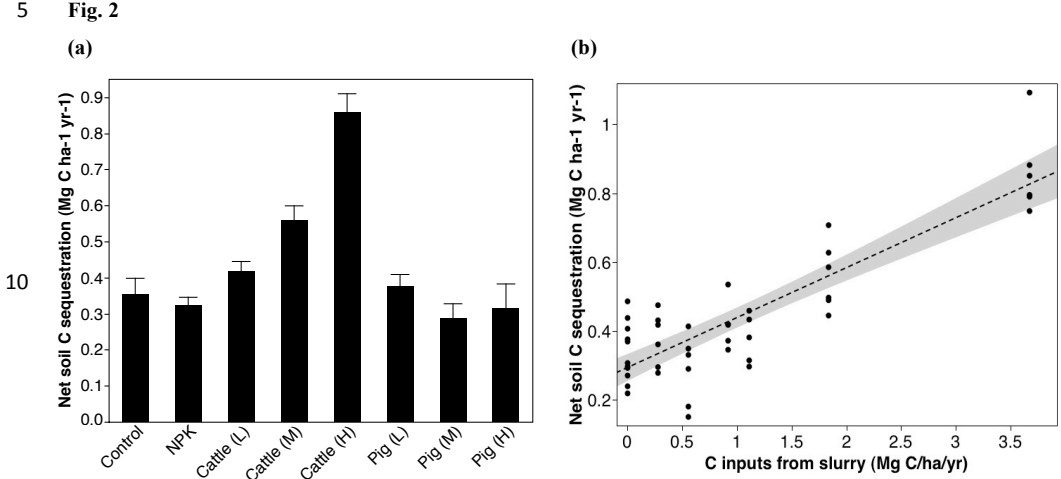





**Fig. 3**

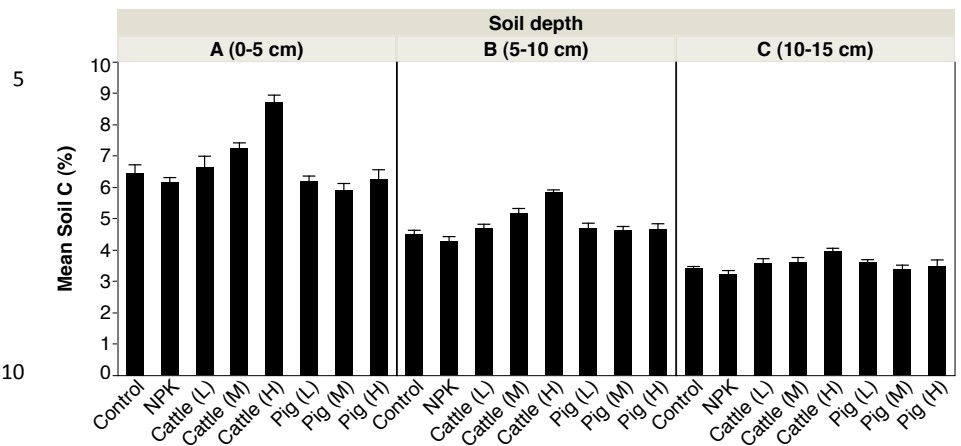

**Fig. 4.**

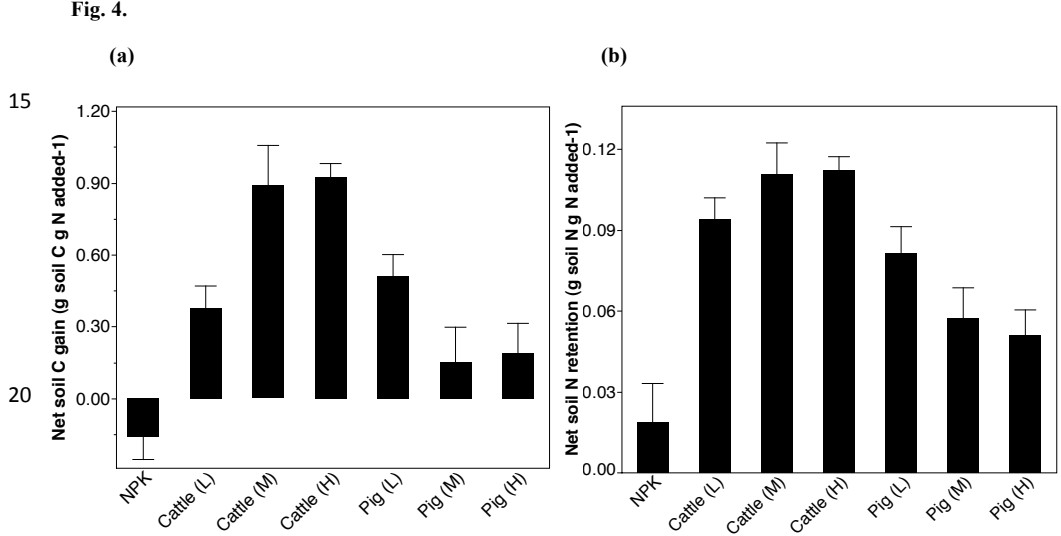



**Fig. 5.**

**(a)**                                               **(b)**

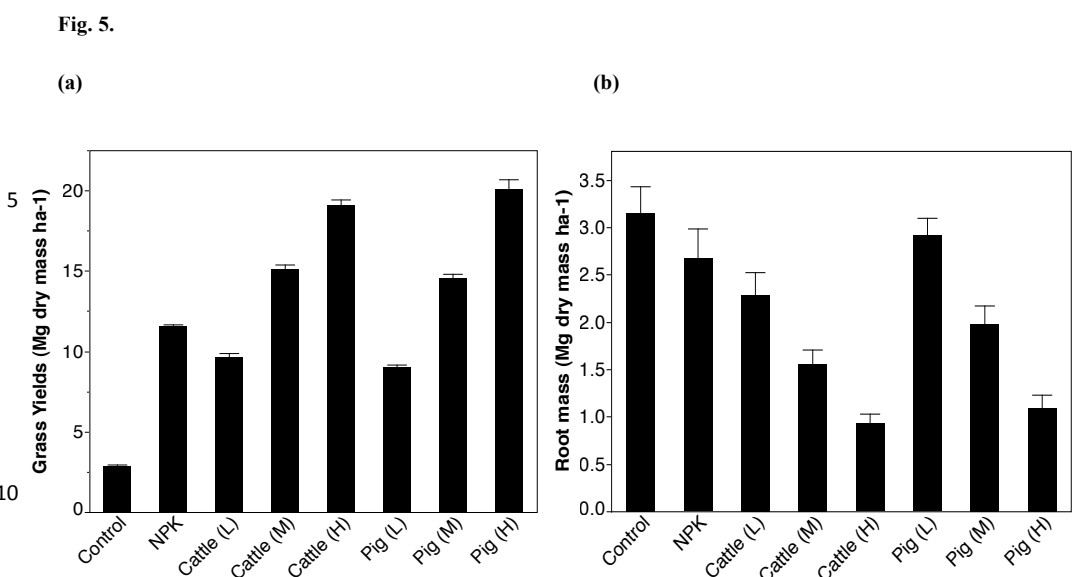

**Fig. 6.**

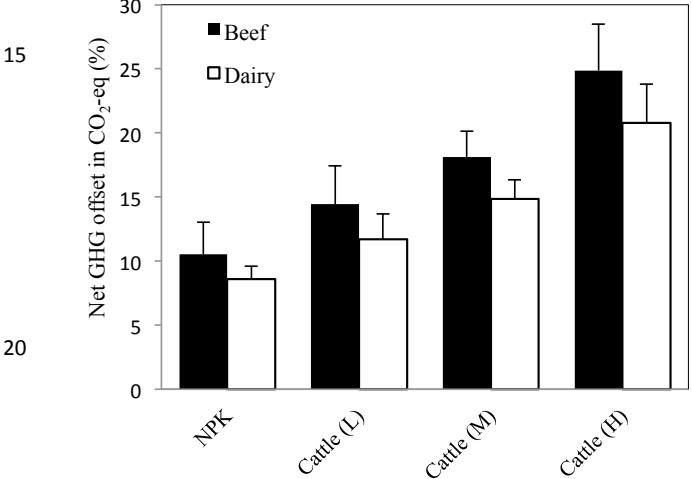