# Peer review of "Long-term nutrient fertilization and the carbon balance of permanent grassland: any evidence for sustainable intensification?"

_Biogeosciences, 2016_

## Referee Comment (RC1) · L. Schipper (Referee) · 2 Jul 2016

Review of: Long-term nutrient fertilization and the carbon balance of permanent grassland: any evidence for sustainable intensification? D. A. Fornara et al.

This paper examines changes in soil C and N in cut pasture systems following application of different pig and cow manures. Control and NPK fertiliser treatments were included. Soil samples were collected to 15 cm. There are few long term trials of this nature and they can provide very valuable information. Like many of these trials, it is not likely that they were primarily established to determine changes in total C stocks and so there can be inevitable short comings. Here the relatively shallow sampling and few bulk density measurements might be criticised. However, I believe it is better to

make the best of the rather unique data that is available for interpretation.

Very interestingly, this study finds increases in soil C and N over 43 years for all treatments including the control (no fertiliser treatment) and the accumulation rates are substantial, as great as 0.86 Mg/ha/y. Establishment of a new carbon stock is reasonably well accepted when there is a major change in land use and management.

The major questions in my mind are: • Why the establishment to a new C stock equilibrium is taking so long particularly in the control treatment (still gaining 0.35 Mg/ha/y) which presumably has been under pasture for some time? • Was there really a difference in C accumulation rate between pig and cattle slurry given these were largely applied at different rates (cattle rates were higher or the same than pig rates). Expanded below. Specific points

1. In the site description, the authors state the site was established in 1970 on an existing sward of ryegrass clover. If it has been in pasture for decades previously but still increasing in soil C stocks this would seem very odd. Was the site cropped in the past and so still recovering from previous C loss? I understand that getting previous land use can be difficult but this is important as the increase in soil C is up to 20% of initial stock in the control soils in ∼40 years (gain of 13 Mg from a base of about 59 Mg).

2. Figure 2b. All the replicates are plotted which presumably gives the tight error bounds, is this reasonable? Error band are not defined in legend.

3. Figure 2b. I am not sure that the authors can assume a linear fit – to me a broken stick model could be fitted that is essentially flat to inputs of about 1.2 Mg C/y and then increases. If correct this could simplify interpretation of why pig and cattle slurry gave different responses. A broken stick model or similar curve would suggest that the first C load of added C is mineralised and the remainder is available for stabilisation. This is an important distinction, the current figure could be interpreted to mean that any addition of external C will build soil C but a broken stick model (or similar model) would

argue that there is a threshold load that is needed before C accumulates. I would have thought the authors need to defend the linear fit. This curve is strongly dependent on the two high C loading from cattle and so the discussion about differences in composition of cattle and pig manure leading to different carbon accumulation (first three paragraphs of the discussion) might more easily be explained by a lower loading rate of pig slurry relative to cattle slurry.

4. The highest pig slurry loading was 1.11 Mg C/ha/y in comparison to lowest cattle slurry application of 0.92 Mg/ha/y both of which had standard errors of about 0.1 (Table 1 – I think these are SE - not stated). Are these significantly different loads? The relative soil C stock changes of 1.05 for pig slurry and 1.09 for cattle slurry with SE of 0.04 and 0.03, respectively (table 1). So for the same slurry load from pigs and cattle gave same amount of C accumulation and no need to try to justify a difference between cattle and pig slurry? Looking at fig 2b the slightly higher C accumulation for the cattle slurry at 0.92 Mg C/ha/y inputs is driven by one high point – the other three points fit within the scatter of the slurry C input of 1.11 Mg C/ha/y. It is important to be clear about this otherwise the reader might conclude that there were indeed differences in C accumulation for cattle and pig slurry when I think this is hardtop justify

5. Figures in general need attention – superscripting missing in some (e.g., fig 4a) and not used in some places (e.g., xaxis fig 2b), different fonts e.g., figure 6. Figure titles don't state what error bars are. Figure 1 what is dashed line – an overall linear fit?

6. Pg 3 (ln 34-36) %C was measured using LECO and then also by muffle furnace – please explain why two different measurements approaches were used– is this to get at inorganic C?

7. Pg 5 ln 7-10. Was the accumulation rate from the control subtracted before calculating Slurry retention factor? I see this is stated in table 1 but missed it in data analysis section.

8. Pg 5 ln 7-10. I guess this assumes that the extra stored C comes only from the slurry

but there was increased plant production also and this has potential to be stabilised in soils also? How was this accounted for? Pg 6 ln 19 states that 16% of slurry C was accumulated in soils but does this ignore the extra pasture production and inputs? But pg 7 ln 36 states 14-15% retention.

9. Pg 7 ln 10 rounding error? should be 10 Mg?

10. Pg 8 ln 31 – I do not understand sentence starting "As opposite . . .

11. Conclusions – I suggest need to tone down the statement "Our findings suggest that permanent grasslands act as a sink rather than source. . ." this may well be true for the current study at present but other have in some cases found losses of C from pasture soils: Schipper et al. (2014), (Meersmans et al., 2009, van Wesemael et al., 2010) for specific soil types. I also think that reference should be made to saturation likely occurring at some stage even with ongoing manure inputs – I think this has been well demonstrated for some of the long-term manure experiments at Rothamsted (e.g.Johnston et al., 2009).

12. I would strongly encourage the authors to provide numeric data rather than just bar graphs, if others want to use this data for comparison or modelling purposes bar graphs are not helpful as you have read off the graphs. Numeric data could be provided in supplementary materials.

References Johnston AE, Poulton PR, Coleman K (2009) Soil organic matter: its importance in sustainable agriculture and carbon dixoide fluxes. In: Advances in Agronomy, Vol 101. (ed Sparks DL) pp 1-57. Meersmans J, Van Wesemael B, De Ridder F, Dotti MF, De Baets S, Van Molle M (2009) Changes in organic carbon distribution with depth in agricultural soils in northern Belgium, 1960-2006. Global Change Biology, 15, 2739-2750. Schipper LA, Parfitt RL, Fraser S, Littler RA, Baisden WT, Ross C (2014) Soil order and grazing management effects on changes in soil C and N in New Zealand pastures. Agriculture Ecosystems & Environment, 184, 67-75. Van Wesemael B, Paustian K, Meersmans J, Goidts E, Barancikova G, Easter M (2010) Agricultural

management explains historic changes in regional soil carbon stocks. Proceedings of the National Academy of Sciences of the United States of America, 107, 14926-14930.

---

## Author Comment (AC1) · 5 Jul 2016

Please see here below our response to the comments provided by Prof. Schipper (Referee).

Review of: Long-term nutrient fertilization and the carbon balance of permanent grassland: any evidence for sustainable intensification? D. A. Fornara et al.

This paper examines changes in soil C and N in cut pasture systems following application of different pig and cow manures. Control and NPK fertiliser treatments were included. Soil samples were collected to 15 cm. There are few long-term trials of this nature and they can provide very valuable information. Like many of these trials, it is
not likely that they were primarily established to determine changes in total C stocks and so there can be inevitable short comings. Here the relatively shallow sampling and few bulk density measurements might be criticised. However, I believe it is better to make the best of the rather unique data that is available for interpretation. Very interestingly, this study finds increases in soil C and N over 43 years for all treatments including the control (no fertiliser treatment) and the accumulation rates are substantial, as great as 0.86 Mg/ha/y. Establishment of a new carbon stock is reasonably well accepted when there is a major change in land use and management.

We really appreciate the very knowledgeable comments provided by Prof. Schipper, which have significantly contributed to improve our manuscript. We agree with the view that long-term grassland experiments are extremely important to understand how the biogeochemistry of these managed ecosystems may be affected by human activities. We also recognize that the application of different management treatments and the addition of more frequent measurements including soil bulk density and other parameters at deeper soil layers might have reduced variability and clarified key trends within our dataset. Nonetheless we would like to highlight the importance of the unique dataset associated with this Long-Term Slurry Experiment, which together with other long-term experiments around the world could contribute to a better mechanistic understanding of how carbon can be lost or gained in permanent grassland soils under intensive management.

The major questions in my mind are: Why the establishment to a new C stock equilibrium is taking so long particularly in the control treatment (still gaining 0.35 Mg/ha/y), which presumably has been under pasture for some time?

Land use previous to the establishment of the Long-Term Slurry Experiment is not known for certain, however, the absence of any specific information suggests that land use was mainly pasture. An important point could be that there had been a 'disturbance' event in 1969 when the pasture was ploughed and reseeded with ryegrass and then the slurry experiment started the following year in 1970. We don't know whether

this disturbance event had created the conditions for accumulating more C in the following years. Our evidence is that C has been accumulating even in recent years after few decades since the disturbance event. We have now added this info on page 3, lines 7-9: "The Long-Term Slurry experiment (LTS) was established in 1970 on a pre-existing sward of perennial ryegrass, which was previously established in 1969 after a ploughing and reseeding event". One of the reasons we think control plots can still show significant rates of C accrual (equal to 0.35 Mg C /ha/y) is due to higher root C pools compared to fertilized plots. High root C pools my provide significant amount of C to soils either via root exudation mechanisms or root decomposition. We have been now planning to measure root decomposition rates in these plots as well as rates of root productivity and microbial community composition to address what factors might influence changes in soil C content through time. Between pages 8-9, we provide this explanation stating that there might be different mechanisms which can lead to more C in soils, either C inputs from animal slurries or C inputs from larger root systems as in the control plots.

Was there really a difference in C accumulation rate between pig and cattle slurry given these were largely applied at different rates (cattle rates were higher or the same than pig rates). This is a good point and we have now clarified it describing what potential differences there might be between pig and cattle slurry applications in determining soil C gains or losses. First, we removed our emphasis on cattle slurry applications as being more beneficial to soil C sequestration than pig slurry applications. Second, we clarify that while all cattle slurry applications were associated with significant changes in soil C accumulation rates when compared to control and NPK plots, pig slurry applications were not. Third, we explain why absolute and relative effects of cattle and pig slurries on soil C sequestration might differ (please see below and also on page 7, section 4.1 of the discussion).

Expanded below. Specific points: 1. In the site description, the authors state the site was established in 1970 on an existing sward of ryegrass clover. If it has been

in pasture for decades previously but still increasing in soil C stocks this would seem very odd. Was the site cropped in the past and so still recovering from previous C loss? I understand that getting previous land use can be difficult but this is important as the increase in soil C is up to 20% of initial stock in the control soils in âĹij40 years (gain of 13 Mg from a base of about 59 Mg). We have now added information about a ploughing and reseeding event occurred in 1969. See page, lines 7-9: "The Long-Term Slurry experiment (LTS) was established in 1970 on a pre-existing sward of perennial ryegrass, which was previously established in 1969 after a ploughing and reseeding event". We agree that net C change in control unfertilized soils has been significant and we don't know for sure whether this is related to potential soil C losses following the reseeding event in 1969. On a parallel study (Carolan R & Fornara DA. 2016. Soil carbon cycling and storage along a chronosequence of re-seeded grasslands: do soil carbon stocks increase with grassland age? Agriculture, Ecosystems and Environment 218, 126–132) we find that reseeding actually determines a short-term increase in CO2 fluxes from soils. If this was the case in our permanent grassland, it has been taking more than 43 years to 'replenish' the soil C pool. I think this potential 'saturation process' deserves more research using long-term experiments.

2. Figure 2b. All the replicates are plotted which presumably gives the tight error bounds, is this reasonable? Error band are not defined in legend. We have changed Fig. 2b following the reviewer indications. We now show mean values and associated standard error bars. We also modified the legend of Fig. 2 as following: "Fig. 2. Relationship between net soil C sequestration rates (Mg C ha-1 yr-1; 0-15 cm depth) and (a) nutrient application treatments, (b) C inputs from animal slurry (Mg C ha-1 yr-1). Experimental treatments same as in Fig. 1. Symbols: Square = no C inputs (control plots), Circles = C inputs from pig slurry, Triangles = C inputs from cattle slurry. Standard errors indicate variation among six replicate values for each treatment".

3. Figure 2b. I am not sure that the authors can assume a linear fit – to me a broken stick model could be fitted that is essentially flat to inputs of about 1.2 Mg C/y and then

increases. If correct this could simplify interpretation of why pig and cattle slurry gave different responses. A broken stick model or similar curve would suggest that the first C load of added C is mineralised and the remainder is available for stabilisation. This is an important distinction, the current figure could be interpreted to mean that any addition of external C will build soil C but a broken stick model (or similar model) would argue that there is a threshold load that is needed before C accumulates. I would have thought the authors need to defend the linear fit. This curve is strongly dependent on the two high C loading from cattle and so the discussion about differences in composition of cattle and pig manure leading to different carbon accumulation (first three paragraphs of the discussion) might more easily be explained by a lower loading rate of pig slurry relative to cattle slurry. We perfectly agree. The linear fit is probably not justified also because pig and cattle slurry applications may return to the soil C compounds with different biochemical properties. We thus prefer not to fit any model but show the different rates of C additions under each slurry type (pig and cattle) and their variation using standard error bars. Note that we have introduced symbols to characterize the two types of slurry in Fig. 2b. We think that the new Fig. 2b fits very well with the three potential explanations (under section 4.1 in the Discussion, pages 7 and 8), that we give to describe why cattle and pig slurry applications may influence soil C sequestration differently.

4. The highest pig slurry loading was 1.11 Mg C/ha/y in comparison to lowest cattle slurry application of 0.92 Mg/ha/y both of which had standard errors of about 0.1 (Table 1 – I think these are SE - not stated). Are these significantly different loads? The relative soil C stock changes of 1.05 for pig slurry and 1.09 for cattle slurry with SE of 0.04 and 0.03, respectively (table 1). So for the same slurry load from pigs and cattle gave same amount of C accumulation and no need to try to justify a difference between cattle and pig slurry? Looking at fig 2b the slightly higher C accumulation for the cattle slurry at 0.92 Mg C/ha/y inputs is driven by one high point – the other three points fit within the scatter of the slurry C input of 1.11 Mg C/ha/y. It is important to be clear about this otherwise the reader might conclude that there were indeed differences in C accumulation for cattle and pig slurry when I think this is hardtop justify. This is a good point indeed and we have now better clarified what potential differences there might be between pig and cattle slurry applications in determining soil C gains or losses. First, we removed our emphasis on cattle slurry applications as being more beneficial to soil C sequestration than pig slurry applications. Second, we clarify that while all cattle slurry applications were associated with significant changes in soil C accumulation rates when compared to control and NPK plots, pig slurry applications were not. Third, we explain why absolute and relative effects of cattle and pig slurries on soil C sequestration might differ. Please see new modified paragraphs on pages 7-8: "Our results show that rates of C accumulation in top-soils significantly increased under high rates of cattle slurry applications. Significant accumulation of C in soils also occurred under low application rates of cattle slurry when compared to control or to NPK nutrient treatments. Instead, pig slurry applications were not associated with significant changes in soil C sequestration rates when compared to unfertilized soils or to soils receiving inorganic (NPK) nutrient additions. Soil C changes in response to pig slurry applications were not as linear as those observed under cattle slurry applications (Fig. 2a, b). We suggest that both absolute and relative effects of cattle vs. pig slurry applications on soil C sequestration will depend on different factors (e.g. animal diets, rates of slurry applications, slurry biochemical composition etc.)  and can thus have different explanations. First, the amount of C added to soils through cattle slurry in our study is higher than in pig slurry (Table 1), and we found a strong positive correlation between C inputs to soils and net soil C accumulation (Fig. 2b). Our estimate is that cattle slurry-C retention efficiency varies between 14-15%, a value that is very similar to a global manure-C retention coefficient of 12% estimated in a recent meta-analysis study (Maillard and Angers, 2014). This means that 85-86% of all C applied yearly to our soils is lost potentially via increased soil $CO_2$ fluxes and/or organic C leaching from soils. However, 15% C retention efficiency associated with high rates of cattle slurry significantly contributed to soil C accrual in this long-term grassland experiment. Second, the biochemical composition of cattle and pig slurries may be significantly different. We did not find any significant difference in total N% (2.9 ± 0.9 vs. 3.1 ± 0.7) or total C% (32.3 ± 2.8 vs. 34.8 ± 3.7) content between pig and cattle slurry, respectively. However, animal ruminants such as cattle are particularly efficient in using C components in their grass feed and excrete high concentrations of slowly digestible organic matter including lignocelluloses (Van Kessel et al., 2000). It is well established that lignin concentrations are negatively related to the biodegradability of organic material in animal manures (Triolo et al., 2011), and that high lignin-to-N ratios tend to slow organic transformations in soils (Chantigny et al., 2002) and can thus lead to greater organic C accumulation in grassland soils (Parton et al., 1987). Recent evidence, however suggests that the concentration of lignin and other biochemical compounds either in animal slurries or in different organic amendments are not necessarily good predictors of their long-term residence time in soils (Lashermes et al., 2009). Thirdly, long-term pig slurry applications were generally associated with poor soil C and N retention per unit of N added (Fig. 4a, b). Here soil C and N retention rates were higher under low rates of pig slurry applications suggesting that increasing application rates of pig slurry may contribute to higher C losses from soils. Previous studies have shown that pig slurry applications can cause soil C losses either by significantly increasing soil $CO_2$ fluxes (Rochette et al., 2000) or by promoting a 'priming effect' whereby high biodegradable pig slurry applications stimulate the mineralization of both native soil C and fresh root-derived material (Angers et al., 2010). Given the high variability in the concentration and abundance of both recalcitrant (e.g. lignin) and labile (e.g. volatile fatty acids) organic compounds in pig and cattle slurries more studies are required to address potential linkages between slurry biochemistry and long-term C and N transformations in soils".

We have also described the meaning of standard error in Table 1. "Effects of C application rates of different animal slurries to soils on (1) Slurry C-retention coefficient, and on (2) Relative soil C stocks change. The 'Slurry-C retention coefficient' represents the average proportion of slurry-C, which has been annually 'retained' in soils after 43 years of slurry applications. The 'Relative soil C stocks change' is the ratio

between SOC stocks in plots receiving slurry and SOC stocks in plots receiving mineral (i.e. NPK) fertilization (sensu Maillard and Angers, 2014). Standard errors indicate variation among six replicate values for each treatment".

5. Figures in general need attention – superscripting missing in some (e.g., fig 4a) and not used in some places (e.g., xaxis fig 2b), different fonts e.g., figure 6. Figure titles don't state what error bars are. Figure 1 what is dashed line – an overall linear fit? We have now edited all figures in the main manuscript as well as figures in the Supplementary material. Superscripts have now been added to title axes, fonts were homogenized across figures. We have also added information on the meaning of our standard errors in all figure legends. Finally we specify that in Fig. 1 the dashed lines represent linear fit functions.

6. Pg 3 (ln 34-36) %C was measured using LECO and then also by muffle furnace – please explain why two different measurements approaches were used– is this to get at inorganic C? Yes, we wanted to assess how much inorganic C% was included in our samples. We have now specified that: "To determine inorganic C concentration, bulk soil subsamples were burned for 16 h at 550 °C in a muffle furnace and ashes analysed for inorganic C and N content" (page 3, lines 36-37).

7. Pg 5 ln 7-10. Was the accumulation rate from the control subtracted before calculating Slurry retention factor? I see this is stated in table 1 but missed it in data analysis section. Yes, this part was not well clarified in our Data Analysis section. We have now added one more sentence to this section: "The difference in total soil C content between each slurry-fertilized plot and the average of the unfertilized controls was also divided by C added over 43 years under each slurry application. We thus calculated the 'Slurry-C retention coefficient' (sensu Maillard and Angers, 2014), which is the average proportion of slurry-C, which has been annually 'retained' in soils after 43 years of slurry applications. Data were analyzed using JMP version 9.0.0 (SAS Institute, Cary, North Carolina, USA)" (page 5, lines 9-13)

8. Pg 5 ln 7-10. I guess this assumes that the extra stored C comes only from the slurry but there was increased plant production also and this has potential to be stabilised in soils also? How was this accounted for? Pg 6 ln 19 states that 16% of slurry C was accumulated in soils but does this ignore the extra pasture production and inputs? But pg 7 ln 36 states 14-15% retention. Yes, here we refer to the ability of these soils to retain C under specific animal slurry applications. It might be that aboveground plant biomass production has contributed over time to additional C inputs to soils but this is very difficult to assess mainly because most of aboveground plant biomass is removed every year during the three annual cuts. Our evidence is that potential C inputs from aboveground plant biomass don't seem to have any significant effect on changes in soil C through time. For example, the productivity of control plots is much lower then NPK fertilized plots (Fig. 5a), however, soil C accumulation seems to be similar or even higher in the control-unfertilized soils (Fig. 1, Fig. 2a). Also high pig slurry applications led to similar or even higher productivity compared to high cattle slurry applications (Fig. 5) but increases in soil C are significantly higher in the cattle slurry fertilized soils. It would be very interesting to measure how much plant litter is returned to soils every year under the different treatments, our impression however is that C inputs from aboveground plant biomass in these managed grasslands could be very limited.

9. Pg 7 ln 10 rounding error? should be 10 Mg? Yes, thanks we have now corrected this. See:" GWP estimated for beef farming ranged from 913 to 952 g $CO_2$-eq m-2 yr-1 (i.e. 9.1 to 9.5 Mg $CO_2$-eq ha-1 yr-1) under increasing cattle slurry applications and 997 g $CO_2$-eq m-2 yr-1 (i.e. 10 Mg $CO_2$-eq ha-1 yr-1) under NPK applications".

10. Pg 8 ln 31 – I do not understand sentence starting "As opposite . . . We agree this is not clear and we simply removed "As opposite" from the sentence. See page 9, line 2.

11. Conclusions – I suggest need to tone down the statement "Our findings suggest that permanent grasslands act as a sink rather than source. . ." this may well be true for the current study at present but other have in some cases found losses of C from

pasture soils:

Schipper et al. (2014), (Meersmans et al., 2009, van Wesemael et al., 2010) for specific soil types. I also think that reference should be made to saturation likely occurring at some stage even with ongoing manure inputs – I think this has been well demonstrated for some of the long-term manure experiments at Rothamsted (e.g.Johnston et al., 2009). We agree with the reviewer and we have now edited this section (pages 9-10). We state: "Our findings suggest that permanent grassland soils may act as sinks of atmospheric $CO_2$ and that an equilibrium state has not been reached after 43 years. However, previous studies show that permanent grassland soils may actually lose C in the long-term (Schipper et al., 2014) and that grassland soils will eventually reach a C saturation point under long-term animal manure applications (Johnston et al., 2009). The soil C accrual observed in our grassland may contribute to reducing the C-footprint of key commodities (i.e. milk)"... We have also added to our reference list two recent studies: Schipper et al., 2014; Johnston et al., 2009.

12. I would strongly encourage the authors to provide numeric data rather than just bar graphs, if others want to use this data for comparison or modelling purposes bar graphs are not helpful as you have read off the graphs. Numeric data could be provided in supplementary materials. We made sure that key rates of soil C and N change through time under different nutrient treatments as well as values of plant productivity and total root mass are reported throughout the Results section. All values are associated with standard error values as well.

References Johnston AE, Poulton PR, Coleman K (2009) Soil organic matter: its importance in sustainable agriculture and carbon dixoide fluxes. In: Advances in Agronomy, Vol 101. (ed Sparks DL) pp 1-57.

Meersmans J, Van Wesemael B, De Ridder F, Dotti MF, De Baets S, Van Molle M (2009) Changes in organic carbon distribution with depth in agricultural soils in northern Belgium, 1960-2006. Global Change Biol- ogy, 15, 2739-2750.

Schipper LA, Parfitt RL, Fraser S, Littler RA, Baisden WT, Ross C (2014) Soil order and grazing management effects on changes in soil C and N in New Zealand pastures. Agriculture Ecosystems & Environment, 184, 67-75.

Van Wesemael B, Paustian K, Meersmans J, Goidts E, Barancikova G, Easter M (2010) Agricultural management explains historic changes in regional soil carbon stocks. Proceedings of the National Academy of Sciences of the United States of America, 107, 14926-14930.

Please also note the supplement to this comment:
http://www.biogeosciences-discuss.net/bg-2016-224/bg-2016-224-AC1-supplement.pdf
* * *
[Figure]

**Fig. 1.** Fig 1

[Figure]

**Fig. 2.** Fig 2a

[Figure]

Fig. 3. Fig 2b

[Figure]

**Fig. 4.** Fig 3

[Figure]

**Fig. 5.** Fig 4a

[Figure]

**Fig. 6.** Fig 4b

[Figure]

**Fig. 7.** Fig 5a

[Figure]

**Fig. 8.** Fig 5b

[Figure]

**Fig. 9.** Fig 6

---

## Referee Comment (RC2) · Anonymous Referee #2 · 8 Jul 2016

The manuscript "Long-term nutrient fertilization and the carbon balance of permanent grassland: any evidence for sustainable intensification? " analyses changes in top soil C stock of 43 years of data from a permanent grassland experiment on organic fertilizer amendment (cattle and pig slurry in different application rates). The manuscript assess key questions such as : how long-term inorganic vs. organic fertilization influences soil C stocks, and how soil C gains (or losses) contribute to the long-term C balance of managed grasslands. The manuscript and data set is interesting and worth to be published. Furthermore the outcomes may give further insight of the effect of C amendment on soil C sequestration of managed grasslands and their latter role to compensate non CO2-farm emissions. However, from the present version needs some

clarification (details on plot experiment ) on serval points (see general comments) and I also encourage authors to look on the data set from different angle: e.g. 2D plot on yield vs. soil C changes (N vs soil C changes) and eventually a 3D (multiple regression) with yield /soil C changes/ N inputs , as this may give further information on slurry amendment thresholds with repect to yield and GHG emissions. . ... Accordingly, I suggest to have (majors) revision before publication

General comments

Introduction I am not sure if authors wanted to avoid the subject or other, but I recommend to cite/mention at some point the equilibrium idea of soil C stocks under grasslands under constant management (Smith et al 2015) (see P2L10-15ff)... authors have mentioned bit and pieces "....it is not clear, however how soil C sequestration will ultimately contribute to the net C balance of intensively managed grasslands.... " ; but I think they can even say that more and more land was used for agriculture in the past 100 year and management was improved over time and barely any management grassland has been managed in the same manner for more than 30yrs.

M&M For reader it's quite difficult to estimate the role of treatments to soil C changes without the necessary information on C_input and N_input (and water?, soil texture??) from slurry and mineral fertiliser. I suggest to add at least a bit more information on the fertilization amendments to the text (P3L14ff), as the volume applied is very vague for slurry and for non farmers the differences between pig and cattle slurry and NPK is difficult to see. Accordingly parts of Tab1-discussion should go to M&M: I had a table in mind like this

Concerning the estimates of soil C stock, personally I found the soil corer of 3cm diameter a bit small to capture spatial heterogeneity. Thus, I missed some information on replicates within a plot, if there were any. Estimates of soil C stock, the description in M&M is a bit shallow. Did authors take into account changes on bulk density over time? (See Stahl et al 2015 but also Ellert et al 2002 SSSAJ) As the soil may have

get compacted or eroded over the 43 years and soil stock analyses should account for that. Accordingly authors should apply a mean BD per layer for all years . . .. Needs to be clarified

Results/Discussion In the section C retention, I suggest to add some estimates on C_outputs vs. C inputs. This will give an idea if slurry amendment (C inputs) could compensated C outputs (removed biomass), and if there is a threshold. For example Cinputs from cattle slurry would approximately correspond to 2 to 8.4 Mg biomass/ha.yr which is half of the harvested biomass (19Mg DM) of cattle(H) and a 10th of the biomass for pig slurry. In the context would be interesting to see 'Slurry-C retention coefficients' (ie C sequestered / C added as slurry) in the context of C removed (output ) from the plot. Besides, this joins my early comment on more information of the N content of the slurry treatments. As there is another point, given that there is little information on N_input reader wonders how biomass production can be similar between cattle and pig slurry? Accordingly , to reader doesn't get evident why authors discussion N_compounds in slurry (eg P8L1-10 may be further developed). Along the same lines, would be nice to see a plot on Yield vs. soil C changes (N vs soil C changes) and eventually a 3D with Yield /soil C changes/ N input . This would give further information on slurry amendment thresholds. And help to provide evidence why pig slurry contributes less to soil C changes (eg P8L13ff). Another 3D (or multiple regression test) to test, may be the yield vs. root biomass vs. soil C changes. As there is a coupled effect (at least what I can interpret from the presented data!). Pig slurry has less C input and higher N_input than cattle slurry (eg Fig 2a and 4b). For control and NPK Ninputs is low and moderate, while C outputs are low and high, respectively Another point the efficiency of the plots yield/Ninput may give and idea on the C sequestration and root growth (i.e. P7L1). The higher the N_input is and the higher there a C outputs the less C is sequestered by plant (and root) litter. This may be compensated by C inputs from slurry! (ie P8L24 may be further developed). Accordingly, the best fertilizer is cattle>pig> ??? either control or NPK (zero C-inputs).( Necessary information is missing for NPK). Suggest to develop a bit further discussion section (P8L25, 30 ff), as there

is little (incomplete) discussion comparing slurry with control and NPK. Nonetheless, difficult to draw general conclusions with top soil analyses. As >30cm fraction may behave differently and loose C over time (eg P9L1ff). How do authors consider the deeper soil layers? Supplementary material Fig 4 and 5 nicely show that in more lower layers (10-15 and 15-30cm) soil C stock is affected by other mechanisms than C and N (in-/out)puts than top soil layer . What authors may add here is the transport of C to deeper soil layers (Rumpel et al.. Kögel-Knabner et al) . . . which may event further increasing the soil C changes.

LCA, though I understand the aim of authors to highlight the importance of soil C sequestration provided by grasslands and the possibility to compensate for non CO2-emissions (see also Soussana et al 2010). I found this part is little developed in the manuscript (mostly in SupplMat). So not sure if there is an added value in the present version. There is also a need for clarification as authors did not mentation the grassland use and how estimates were done. Or did authors only account for cut grassland. (Needs to be clarified) . For ever estimate one would need animal stocking rates /ha. Grazing /barn period to estimate partition between manure and dejection emissions in the field, etc. The same for supplements of concentrates to diet »>Accordingly for reader doesn't get clear what is the part of each of the mentioned emissions sources : (3) enteric fermentation of ruminant livestock, (4) manure management (CH$_4$), (5) manure management (N$_2$O-N), (6) managed soils (CH$_4$), (7) managed soils (N$_2$O-N), (8) milk yields, (9) beef yields, (10) feed concentrate production and transportation, (11) fertilizer production, and (12) machinery use. Anyhow to improve the section and readers attention I suggest to add GHG emissions in in Fig 6 in order to better capture the "offset idea".

Specific comments

Table 1 add NPK and control

P2L15 guess you have to cite Smith et al 2015 at some point Authors may even say that

more and more land was used for agriculture in the past 100 year and management was improved over time

P2L20 authors may even cite Aneja et al 2008 showing the fertilizer procution and consumption since the 60s P3L14ff suggest to add a bit more information of the fertilization amendments to the text, as the volume applied is very vague for slurry. Suggest to add at least some proxies of N and C to the L, M, H amendements.

P4L12 suggest to precise how estimates dealt with grassland use (grazing/cut)

P6L38 what is the yield of NPK??? Should be in between slurry and control??!!

P7L 22ff suggest to better clarifies the sentences otherwise the conclusion can become mis leading The application of high rates of cattle slurry (i.e. XX % of Cinputs Nfertilisation) significantly contributed to increase soil C sequestration up to 0.86 Mg C ha -1 yr -1 (86 g C m -2 yr - 1), whereas the application of pig slurry and inorganic NPK fertilizer with lower or non C input did not have any significant C sequestration benefit when compared to unfertilized (control) soils.

L26 ff idem to above "soils may can act as C sinks but when C losses from frequent grass mowing are compensated primary productivity (ie results of N fertilization) and organic C inputs (i.e. slurry)

P7L32 "Our results provide evidence that, under comparable biomass production, cattle slurry applications are more effective than pig slurry applications in terms of top soil C accumulation. This this positive effect could have multiple explanations.

P8L2 "We did not find any significant difference in total N% (2.9 ± 0.9 vs. 3.1 ± 0.7) or total C% (32.3 ± 2.8 vs. 34.8 ± 3.7)" suggest to move information also to M&M section

P2L31ff. "…As opposite in the control plots which do not receive any C and N addition, soil C accumulation may be partly explained by low C outputs via harvests and higher root mass production (possibly resulting from positive grass-legume interactions), which contributes to higher C inputs to soils when compared to the reduced root

systems of fertilized slurry plots (see Fig. 5b).

P9L20 "...to beef management simply because dairy cattle are associated with higher CH4 emissions from enteric fermentation and manure management." Suppose that dairy cattle has also higher emissions due to the fact that for milking they stay more in the barn and less on the paddock as beef cows. IE barn emissions (manure, etc ) are higher for dairy cows

---

## Referee Comment (RC3) · L. Schipper (Referee) · 9 Jul 2016

Thanks for the responses.

In conclusions I dont agree that the previous work that I did suggests in the long-term pasture soils lose carbon. I think there needs to be a broader discussion here 'some' pasture systems will gain C and some will lose with time and many/most will likely be at steady state. Our previous work (Schipper et al 2014) demonstrated that the majority of soils in long-term pasture on flat land in New Zealand appear at steady state except for a couple of soil orders Gley (losses attributed to drainage and enhanced C respiration, as was suggested in Belgium) and Allophanic soils (we do not have an explanation for this loss).

In contrast pasture on hill country was gaining C, possibly due to recovery after past sheet wash erosion but not known.

So in the end we we need to understand the variety of pasture management systems that may increase or decrease soil C but a general statement that carbon increases or decreases in pastures is not really supported, it depends strongly on past land use and current management.

---

## Author Comment (AC2) · 19 Jul 2016

Please see here below in the attached PDF document our response to Referee 2.
The manuscript "Long-term nutrient fertilization and the carbon balance of permanent grassland: any evidence for sustainable intensification? " analyses changes in top soil C stock of 43 years of data from a permanent grassland experiment on organic fertilizer amendment (cattle and pig slurry in different application rates). The manuscript

assess key questions such as: how long-term inorganic vs. organic fertilization influences soil C stocks, and how soil C gains (or losses) contribute to the long-term C balance of managed grasslands. The manuscript and data set is interesting and worth to be published. Furthermore the outcomes may give further insight of the effect of C amendment on soil C sequestration of managed grasslands and their latter role to compensate non $CO_2$-farm emissions. However, from the present version needs some clarification (details on plot experiment ) on serval points (see general comments) and I also encourage authors to look on the data set from different angle: e.g. 2D plot on yield vs. soil C changes (N vs soil C changes) and eventually a 3D (multiple regression) with yield /soil C changes/ N inputs , as this may give further information on slurry amendment thresholds with repect to yield and GHG emissions. Accordingly, I suggest to have (majors) revision before publication.

We appreciate the thoughtful comments of the reviewer, which we found very useful and helped us to further improve our manuscript. We have now added more information on N inputs from animal slurry. We agree with the view that there might be different, alternative ways to show the combined effects of multiple predictor factors. Here below we describe how we produced new graphs while searching for potential relationships between yields and changes in soil C. However, one aspect that we want to clarify immediately is that there is not any significant relationship between plant yields and changes in soil C ($R^2$=0.05, $F_{1,48}$ = 2.33, P = 0.13, linear regression). The most important variable affecting net soil C changes is C inputs from slurry (Fig. 2b) as shown in the manuscript. Please see here below our responses to all points raised by the reviewer.

Please also note the supplement to this comment:
http://www.biogeosciences-discuss.net/bg-2016-224/bg-2016-224-AC2-supplement.pdf

─────────────────────

**Supplement:**

**The manuscript "Long-term nutrient fertilization and the carbon balance of permanent grassland: any evidence for sustainable intensification? " analyses changes in top soil C stock of 43 years of data from a permanent grassland experiment on organic fertilizer amendment (cattle and pig slurry in different application rates). The manuscript assess key questions such as: how long-term inorganic vs. organic fertilization influences soil C stocks, and how soil C gains (or losses) contribute to the long-term C balance of managed grasslands. The manuscript and data set is interesting and worth to be published. Furthermore the outcomes may give further insight of the effect of C amendment on soil C sequestration of managed grasslands and their latter role to compensate non CO2-farm emissions. However, from the present version needs some clarification (details on plot experiment ) on serval points (see general comments) and I also encourage authors to look on the data set from different angle: e.g. 2D plot on yield vs. soil C changes (N vs soil C changes) and eventually a 3D (multiple regression) with yield /soil C changes/ N inputs , as this may give further information on slurry amendment thresholds with repect to yield and GHG emissions. Accordingly, I suggest to have (majors) revision before publication.**

We appreciate the thoughtful comments of the reviewer, which we found very useful and helped us to further improve our manuscript. We have now added more information on N inputs from animal slurry. We agree with the view that there might be different, alternative ways to show the combined effects of multiple predictor factors. Here below we describe how we produced new graphs while searching for potential relationships between yields and changes in soil C. However, one aspect that we want to clarify immediately is that there is not any significant relationship between plant yields and changes in soil C ($R^2$=0.05, $F_{1,48}$ = 2.33, P = 0.13, linear regression). The most important variable affecting net soil C changes is C inputs from slurry (Fig. 2b) as shown in the manuscript. Please see here below our responses to all points raised by the reviewer.

**General comments**
**Introduction I am not sure if authors wanted to avoid the subject or other, but I recommend to cite/mention at some point the equilibrium idea of soil C stocks under grasslands under constant management (Smith et al 2015) (see P2L10-15ff): : : authors have mentioned bit and pieces ": : :.it is not clear, however how soil C sequestration will ultimately contribute to the net C balance of intensively managed grasslands: : :. " ; but I think they can even say that more and more land was used for agriculture in the past 100 year and management was improved over time and barely any management grassland has been managed in the same manner for more than 30yrs.**

We agree with the reviewer and we have now addressed the soil C equilibrium issue in managed

grasslands. On page 2 (lines 12-20) we have added three more sentences and identified knowledge gaps, which are partly addressed by our study. We state: "Altogether these studies suggest that grassland soils under human management can potentially act as significant C sinks. It is likely however that grassland soils under constant management and environmental conditions will eventually reach relatively stable C levels. Using data from long-term soil chronosequences Smith (2014) suggests that soil C stocks may reach a relatively stable equilibrium around 100 years after a land use change. Despite soils can still accumulate C after decades of human management it is not clear how soil C content will change under long-term nutrient applications (>40 years). It is not clear either how soil C sequestration will contribute to the net C balance of intensively managed grasslands especially when we account for large GHG emissions from the livestock sector (i.e. emissions from enteric fermentation of ruminant livestock, manure management etc.)".

**M&M For reader its quite difficult to estimate the role of treatments to soil C changes without the necessary information on C_input and N_input (and water?, soil texture??) from slurry and mineral fertiliser. I suggest to add at least a bit more information on the fertilization amendments to the text (P3L14ff), as the volume applied is very vague for slurry and for non farmers the differences between pig and cattle slurry and NPK is difficult to see. Accordingly parts of Tab1-discussion should go to M&M: I had a table in mind like this.**
We agree with the reviewer and made sure this key information is now available in the Material and Methods section. For example on page 3, line 16-17 is specified that: "The soil is a clay loam (42% sand, 24% silt and 34% clay) overlying Silurian shale and greywacke (Christie, 1987)". On page 3 lines 26-28, we also clarify that: "The N content of animal slurry as averaged for 10 recent years is 0.16, 0.32 and 0.64 Mg N ha$^{-1}$ yr$^{-1}$ for Cattle (L), (M) and (H) respectively, while pig slurry applications (L), (M) and (H) contributed to 0.13, 0.27 and 0.54 Mg N ha$^{-1}$ yr$^{-1}$ being added to the soil". On lines 28-29 we state that: "We also estimated C inputs from animal slurries, which are summarized in Table 1".

**Concerning the estimates of soil C stock, personally I found the soil corer of 3cm diameter a bit small to capture spatial heterogeneity. Thus, I missed some information on replicates within a plot, if there were any. Estimates of soil C stock, the description in M&M is a bit shallow. Did authors take into account changes on bulk density over time? (See Stahl et al 2015 but also Ellert et al 2002 SSSAJ) As the soil may have get compacted or eroded over the 43 years and soil stock analyses should account for that. Accordingly authors should apply a mean BD per layer for all years. Needs to be clarified.**
On page 4 lines 1-3 we specify that: "Soil samples were collected in 1972, 1975, 1986, 1988, 1991, 1996, 2001, 2007, 2012 and 2013 at three depth-intervals (0-5, 5-10, 10-15 cm) and at 8 random locations within each experimental plot (29.75 m$^2$ each) in February each year using a 3 cm diameter soil corer". This means that we have collected 24 soil samples within each plot (8 samples for each soil depth level, thus 24*48 plots = 1,152 per year and 11,420 for the 10 years used in this study. On page 4, lines 9-13 we specify that: "Two larger soil cores were collected in each plot at two different soil depths (0-10 and 10-20 cm) to measure soil bulk density as the ratio between air-dried (stone-free) soil and soil volume. Soil bulk density was measured in 1970

on a grassland field beside the LTS experimental site and re-measured in 2013 in each LTS experimental plot". On page 5, lines 26-30 we clarify that our evidence is that: "...soil bulk density averaged between 0-10 cm depth was very similar across nutrient treatments with values ranging between 1.027 and 1.039 g cm$^{-3}$ thus <1% variation across our soils after 43 years of different nutrient applications. Similarly, soil bulk density measured between 10-20 cm ranged between 1.025 and 1.031 g cm$^{-3}$ (<1% variation across nutrient treatments). Our reference value for soil bulk density in 1970 was equal to 1.1, a value, which was used to calculate soil C stocks for 1970 and 1975". Based on our measurements these soils have had very little (or at least negligible) changes in soil bulk density either between 1970 and 2013 or across the various nutrient treatments.

**Results/Discussion In the section C retention, I suggest to add some estimates on C_outputs vs. C inputs. This will give an idea if slurry amendment (C inputs) could compensated C outputs (removed biomass), and if there is a threshold. For example C inputs from cattle slurry would approximately correspond to 2 to 8.4 Mg biomass/ha.yr which is half of the harvested biomass (19Mg DM) of cattle (H) and a 10th of the biomass for pig slurry. In the context would be interesting to see 'Slurry-C retention coefficients' (ie C sequestered / C added as slurry) in the context of C removed (output ) from the plot. Besides, this joins my early comment on more information of the N content of the slurry treatments. As there is another point, given that there is little information on N_input reader wonders how biomass production can be similar between cattle and pig slurry? Accordingly , to reader doesn't get evident why authors discussion N_compounds in slurry (eg P8L1-10 may be further developed).**

We understand here why the reviewer is focusing on plant yields and on 'C outputs' through plant biomass removal. Indeed, a significant amount of C is 'yielded' and removed every year after the three cuts (see Fig. 5a). However, the C removed in plant biomass (yields) is not linked to changes in soil C content and pools in the long-term grassland experiment. In terms of C pools, the aboveground plant C pool does not contribute to new C inputs to soils (as plant detritus) because all this plant biomass is removed from the plots. In terms of long-term changes in the C balance of this permanent grassland, plant C associated with biomass yields is transformed and 'converted' into animal biomass and animal wastes (solids, liquids and gaseous-related losses), whose contributions to net C balance are then taken into account through various estimates (e.g. enteric fermentation, manure management for $CH_4$ and for $N_2O$-N), managed soils for $CH_4$ and $N_2O$-N, milk yields, beef yields etc.).

We have calculated here below the ratio between C inputs from slurries and C outputs through plant biomass removal, which could be defined as the 'C use efficiency' of this particular farming system. The graph below simply shows that plots fertilized with cow slurries receive between 0.2 to 0.4 times the C that is removed every year through plant biomass harvest. A significant pool of the plant C harvested has however been converted into animal biomass and thus animal wastes which along different pathways contribute to GHG emissions. We think that the graph below might have some useful information for agronomic purposes in relation to how slurry identity (cattle vs. pig) might affect yields or how this particular farming system recycles C from fresh plant biomass to slurry applications. This however is less relevant to the question related to how long-term nutrient management influences the soil C balance of these grasslands.

[Figure]

We have also calculated the ratio between the amount of N applied through slurries and C outputs via plant biomass removal, which could give indications of how N use efficiency associated with different animal slurries might influence plant yields. See here below:

[Figure]

Again this information could be used for agronomic purposes to understand how cattle and pig slurry might be more or less beneficial to plant yields.

**Along the same lines, would be nice to see a plot on Yield vs. soil C changes (N vs soil C changes) and eventually a 3D with Yield /soil C changes/ N input . This would give further information on slurry amendment thresholds. And help to provide evidence why pig slurry contributes less to soil C changes (eg P8L13ff). Another 3D (or multiple regression test) to test, may be the yield vs. root biomass vs. soil C changes. As there is a coupled effect (at least what I can interpret from the presented data!). Pig slurry has less C input and higher N_input than cattle slurry (eg Fig 2a and 4b). For control and NPK N inputs is low and moderate, while C outputs are low and high, respectively Another point the efficiency of**

**the plots yield/Ninput may give and idea on the C sequestration and root growth (i.e. P7L1). The higher the N_input is and the higher there a C outputs the less C is sequestered by plant (and root) litter. This may be compensated by C inputs from slurry! (ie P8L24 may be further developed). Accordingly, the best fertilizer is cattle> pig> ??? either control or NPK (zero C-inputs).( Necessary information is missing for NPK).**

The reviewer suggests searching for potential linkages between yields, soil C changes and N inputs from slurries perhaps combining these 3 factors in a 3D graph. We think that addressing potential relationships between soils' ability to gain C per unit of N added is very important and we indeed show these relationships in Fig. 4a, b. These figures show for example how the two high cattle slurry applications are associated with higher C gains per unit of N added. We think though that by adding an extra variable it makes it more complicated to understand and interpret potential relationships between mutliple underlying biogeochemical processes. For example, we produced a 3D graph below as the reviewer suggested where we also added the yield information. Here it is difficult to interpret how the three variables might influence soil C. This is partly because there are complex biogeochemical processes that interact to affect N use efficiency, soil C accumulation and plant biomass production and partly because yields and soil C sequestration may not be related in these agricultural grasslands where most of aboveground plant biomass is removed and thus C inputs from plant detritus to soils remain very low. In a multiple regression analysis where C inputs from slurries and yields are added as predictor variables we found that plant yields *per se'* do not explain changes in soil C sequestration ($F_{1,47}$ = 3.36, P = 0.07) whereas C inputs from slurries have a strong effect on changes in soil C ($F_{1,47}$ = 95.2, P < 0.0001).

[Figure]

We think that in order to understand why cattle and pig slurry applications might influence soil C sequestration in different ways we need to look at the biochemical composition of these slurries and the effects they might have on soil biogeochemistry. Now we have modified this section of the Discussion (see page 8, lines 5-39): "Our results show that rates of C accumulation in topsoils significantly increased under high rates of cattle slurry applications. Significant accumulation of C in soils also occurred under low application rates of cattle slurry when compared to control or to NPK nutrient treatments. Instead, pig slurry applications were not associated with significant changes in soil C sequestration rates when compared to unfertilized soils or to soils receiving inorganic (NPK) nutrient additions. Soil C changes in response to pig slurry applications were not as linear as those observed under cattle slurry applications (Fig. 2a, b). We suggest that both absolute and relative effects of cattle *vs.* pig slurry applications on soil C sequestration will depend on different factors (e.g. animal diets, rates of slurry applications, slurry biochemical composition etc.) and can thus have different explanations.

First, the amount of C added to soils through cattle slurry in our study is higher than in pig slurry (Table 1), and we found a strong positive correlation between C inputs to soils and net soil C accumulation (Fig. 2b). Our estimate is that cattle slurry-C retention efficiency varies between 14-15%, a value that is very similar to a global manure-C retention coefficient of 12% estimated in a recent meta-analysis study (Maillard and Angers, 2014). This means that 85-86% of all C applied yearly to our soils is lost potentially via increased soil $CO_2$ fluxes and/or organic C leaching from soils. However, 15% C retention efficiency associated with high rates of cattle slurry significantly contributed to soil C accrual in this long-term grassland experiment.

Second, the biochemical composition of cattle and pig slurries may be significantly different. We did not find any significant difference in total N% (2.9 ± 0.9 vs. 3.1 ± 0.7) or total C% (32.3 ± 2.8 vs. 34.8 ± 3.7) content between pig and cattle slurry, respectively. However, animal ruminants such as cattle are particularly efficient in using C components in their grass feed and excrete high concentrations of slowly digestible organic matter including lignocelluloses (Van Kessel et al., 2000). It is well established that lignin concentrations are negatively related to the biodegradability of organic material in animal manures (Triolo et al., 2011), and that high lignin-to-N ratios tend to slow organic transformations in soils (Chantigny et al., 2002) and can thus lead to greater organic C accumulation in grassland soils (Parton et al., 1987). Recent evidence, however suggests that the concentration of lignin and other biochemical compounds either in animal slurries or in different organic amendments are not necessarily good predictors of their long-term residence time in soils (Lashermes et al., 2009).

Thirdly, long-term pig slurry applications were generally associated with poor soil C and N retention per unit of N added (Fig. 4a, b). Here soil C and N retention rates were higher under low rates of pig slurry applications suggesting that increasing application rates of pig slurry may contribute to higher C losses from soils. Previous studies have shown that pig slurry applications can cause soil C losses either by significantly increasing soil $CO_2$ fluxes (Rochette et al., 2000) or by promoting a 'priming effect' whereby high biodegradable pig slurry applications stimulate the mineralization of both native soil C and fresh root-derived material (Angers et al., 2010). Given the high variability in the concentration and abundance of both recalcitrant (e.g. lignin) and labile (e.g. volatile fatty acids) organic compounds in pig and cattle slurries more studies are required to address potential linkages between slurry biochemistry and long-term C and N transformations in soils".

**Suggest to develop a bit further discussion section (P8L25, 30 ff), as there is little (incomplete) discussion comparing slurry with control and NPK. Nonetheless, difficult to draw general conclusions with top soil analyses. As >30cm fraction may behave differently**

**and loose C over time (eg P9L1ff). How do authors consider the deeper soil layers? Supplementary material Fig 4 and 5 nicely show that in more lower layers (10-15 and 15-30cm) soil C stock is affected by other mechanisms than C and N (in-/out)puts than top soil layer . What authors may add here is the transport of C to deeper soil layers (Rumpel et al.. Kögel-Knabner et al) : : : which may event further increasing the soil C changes.**

We have now modified section 4.1 of our discussion (page 8) and on page 9 (lines 1-14) we discuss why NPK fertilization may not have significant effects on soil C: "In the current study, cattle slurry led to greater C and N soil retention rates per unit of N added, than inorganic synthetic fertilizer. Soils receiving high rates of cattle slurry retained 0.89-0.94 grams of C per gram of N added (Fig. 4a) whereas soils receiving NPK additions did not retain any C per unit of N added. This may explain why NPK-fertilized soils did not show any significant change in C stocks when compared to the unfertilized control. The results that long-term NPK fertilization does not have any additional benefit for soil C sequestration is supported by previous findings (Fornara et al., 2013), which show that despite chronic multi-nutrient additions can greatly contribute to increased grass yields, they significantly decreased total root mass and did not have any effect on soil C stocks compared to unfertilized soils. Our study also suggests that different processes might drive long-term soil C accumulation, for example, in soils amended with cattle slurry it is clear that C accumulation is strongly increased by 'external' C additions through liquid manure. In the control plots which do not receive any C addition, soil C accumulation may be partly explained by higher root mass (possibly resulting from positive grass-legume interactions), which contributes to higher C inputs to soils when compared to the reduced root systems of fertilized plots (see Fig. 5b)".

We agree with the reviewer that there might be C changes in deeper soil layers. Our evidence is that most C changes occur in the 0-15 cm soil layer and become not significant in the 15-30 cm soil layer. We think it is worth in future studies sampling soils down to 100 cm depth to address whether there has been any C movement to deeper soils.

**LCA, though I understand the aim of authors to highlight the importance of soil C sequestration provided by grasslands and the possibility to compensate for non CO2-emissions (see also Soussana et al 2010). I found this part is little developed in the manuscript (mostly in SupplMat). So not sure if there is an added value in the present version. There is also a need for clarification as authors did not mentation the grassland use and how estimates were done. Or did authors only account for cut grassland. (Needs to be clarified) . For ever estimate one would need animal stocking rates /ha.**

Our aim here is to estimate how intensively used agricultural grasslands (3 cuts and 3 nutrient fertilization applications) might contribute to global change when accounting for soils' ability to sequester C in the long-term. In the supplementary material we specify that we assume a livestock stocking rate of 2 animals per hectare. Please see page 4, lines 1-14: "To estimate the long-term net C balance of our permanent grassland we calculated the Global Warming Potential (GWP) associated with each experimental nutrient treatment (Robertson et al., 2000; Fornara et al., 2011). Net changes in soil $CO_2$ sequestration were calculated between 1970 and 2013 in the 0-15 cm soil depth layer. We are aware that *IPCC Guidelines* define soil carbon stocks as organic carbon incorporated into mineral soil horizons to a depth of 30 cm. In our study, we have measured soil C stocks between 15 and 30 cm depth in 1998 and 2013 (see Supplementary Fig.

5); we did not find, however, any significant difference in C stocks between nutrient treatments. We observed significant changes in soil C content and stocks only in the 0-15 cm soil depth and we thus used these rates of soil C sequestration in our GWP calculations. All GHG emissions associated with the management of our permanent grassland were calculated using information from IPCC reports (Dong et al., 2006; De Klein et al., 2006; Myhre et al., 2013) and from multiple peer-reviewed scientific papers. GHG emissions were converted to $CO_2$ equivalents ($CO_2$-eq) assuming a 100 year time horizon (Myhre et al., 2013). To simulate grassland management intensification we assumed a cattle-stocking rate of 2 animals (i.e. livestock units) per hectare".

**Grazing /barn period to estimate partition between manure and dejection emissions in the field, etc. The same for supplements of concentrates to diet»>Accordingly forreader doesn't get clear what is the part of each of the mentioned emissions sources: (3) enteric fermentation of ruminant livestock, (4) manure management (CH 4 ), (5) manure management (N 2 O-N), (6) managed soils (CH 4 ), (7) managed soils (N 2 ON), (8) milk yields, (9) beef yields, (10) feed concentrate production and transportation, (11) fertilizer production, and (12) machinery use. Anyhow to improve the section and readers attention I suggest to add GHG emissions in in Fig 6 in order to better capture the "offset idea".**
We agree with the reviewer and we have now added an extra figure (Fig. 6a) where we show overall GHG emissions and then the soil C offset ability in Fig. 6b. We have also modified the discussion (see page 9 line 30 to page 10 line 7): "We estimated that overall GHG emissions from permanent grassland ranged between 10.7 to 15.2 Mg $CO_{2-eq}$ ha$^{-1}$ yr$^{-1}$ (Fig. 6a). Our evidence is that long-term $CO_2$ sequestration in grassland soils can offset between ~9% and ~25% of overall GHG emissions estimated at the 'farm-gate' for dairy and beef farms (Fig. 6b). Dairy cattle management at the farm-level is less able to offset GHGs when compared to beef management simply because dairy cattle are associated with higher $CH_4$ emissions from enteric fermentation and manure management. Our estimates of soils' ability to offset overall GHG emissions agree with average values obtained by the carbon-foot print calculator 'BovIs' developed by the Agri-Food Biosciences Institute (see Aubry et al., 2013), validated by comparing GHG emissions from a hundred farms across Northern Ireland. Our results show relatively high GWP values associated with farm management suggesting that these livestock systems contribute to global climate change by emitting between 9 and 12 Mg $CO_{2-eq}$ ha$^{-1}$ yr$^{-1}$. We are not aware of any other study, which has quantified net C balance of permanent grassland under long-term inorganic and organic nutrient fertilization. However, our estimates are comparable with the range of net emissions (i.e. GWPs) of 7.5-to-11.2 Mg $CO_{2-eq}$ ha$^{-1}$ yr$^{-1}$, associated with intensively managed permanent grasslands in the Allgäu region of Germany (Haas et al., 2001). Moreover, if we consider GWP values in relation to the production of an important agricultural commodity (i.e. milk) we estimate that the C-footprint of milk production in our grassland ranges between 0.96 to 1.03 Mg $CO_{2-eq}$/ Mg milk, which is similar to average values estimated from across many farms under similar environmental conditions (DairyCo, 2014; O'Brien et al., 2014)".

**Specific comments**

**Table 1 add NPK and control**

Table 1 shows two parameters in particular: (1) Slurry C-retention coefficient, and the (2) Relative soil C stocks change under different animal slurry applications. The 'Slurry-C retention coefficient' represents the average proportion of slurry-C, which has been annually 'retained' in soils after 43 years of slurry applications. The 'Relative soil C stocks change' is the ratio between SOC stocks in plots receiving slurry and SOC stocks in plots receiving mineral (i.e. NPK) fertilization. The relationships between control and NPK with soil C are shown in Figs 1, 2, 3 and 4 (NPK).

**P2L15 guess you have to cite Smith et al 2015 at some point Authors may even say that more and more land was used for agriculture in the past 100 year and management was improved over time.**

We have now changed this section (page 2, lines 14-20) where we state that: "Using data from long-term soil chronosequences Smith (2014) suggests that soil C stocks may reach a relatively stable equilibrium around 100 years after a land use change. Despite soils can still accumulate C after decades of human management it is not clear how soil C content will change under long-term nutrient applications (>40 years). It is not clear either how soil C sequestration will contribute to the net C balance of intensively managed grasslands especially when we account for large GHG emissions from the livestock sector (i.e. emissions from enteric fermentation of ruminant livestock, manure management etc.).

**P2L20 authors may even cite Aneja et al 2008 showing the fertilizer procution and consumption since the 60s P3L14ff suggest to add a bit more information of the fertilization amendments to the text, as the volume applied is very vague for slurry. Suggest to add at least some proxies of N and C to the L, M, H amendements.**

We agree with the reviewer and we have now added this information in the Methods section (page 3, lines 26-29). "The N content of animal slurry as averaged for 10 recent years is 0.16, 0.32 and 0.64 Mg N ha$^{-1}$ yr$^{-1}$ for Cattle (L), (M) and (H) respectively, while pig slurry applications (L), (M) and (H) contributed to 0.13, 0.27 and 0.54 Mg N ha$^{-1}$ yr$^{-1}$ being added to the soil. We also estimated C inputs from animal slurries, which are summarized in Table 1".

**P4L12 suggest to precise how estimates dealt with grassland use (grazing/cut)**

On page 4, lines 18-21 we specify that: "Grass biomass was harvested 3 times every year (usually in May, July and September). Each experimental plot has a harvestable area of 18.75 m$^2$ within which all grass biomass is cut at a height of 5 cm with a plot harvester. Total fresh plant biomass is weighed in the field at each harvest and a sub-sample of 300 g fresh weight is then oven dried at 80°C to estimate dry matter production per hectare per year under each nutrient treatment".

**P6L38 what is the yield of NPK??? Should be in between slurry and control??!!**

We have now added this information on page 7, lines 11-13: "Plant yields averaged 11.5 ± 0.35 Mg dry mass ha$^{-1}$ under NPK fertilization whereas yields were significantly lower in the control plots being 2.8 ± 0.01 Mg dry mass ha$^{-1}$; Fig. 5a".

**P7L 22ff suggest to better clarifies the sentences otherwise the conclusion can become misleading. The application of high rates of cattle slurry (i.e. XX % of C inputs N fertilisation) significantly contributed to increase soil C sequestration up to 0.86 Mg C ha -1 yr -1 (86 g C m -2 yr - 1), whereas the application of pig slurry and inorganic NPK fertilizer with lower or non C input did not have any significant C sequestration benefit when compared to unfertilized (control) soils.**

We agree with the reviewer and we have now edited this section, please see page 8, lines 5-13: "Our results show that rates of C accumulation in top-soils significantly increased under high rates of cattle slurry applications. Significant accumulation of C in soils also occurred under low application rates of cattle slurry when compared to control or to NPK nutrient treatments. Instead, pig slurry applications were not associated with significant changes in soil C sequestration rates when compared to unfertilized soils or to soils receiving inorganic (NPK) nutrient additions. Soil C changes in response to pig slurry applications were not as linear as those observed under cattle slurry applications (Fig. 2a, b). We suggest that both absolute and relative effects of cattle *vs.* pig slurry applications on soil C sequestration will depend on different factors (e.g. animal diets, rates of slurry applications, slurry biochemical composition etc.) and can thus have different explanations".

**L26 ff idem to above "soils may can act as C sinks but when C losses from frequent grass mowing are compensated primary productivity (ie results of N fertilization) and organic C inputs (i.e. slurry)**

Not sure if we completely understand this point raised by the reviewer. We think the point to clarify here is that plant C removed after the three cuts is not directly related to net C changes in soils. Grass yields between cattle, pig and NPK treatments are comparable but cattle-fertilized plots tend to accumulate more C in soils in the long-term compared to NPK or pig slurries.

**P7L32 "Our results provide evidence that, under comparable biomass production, cattle slurry applications are more effective than pig slurry applications in terms of top soil C accumulation. This this positive effect could have multiple explanations.**

We have now modified this section, please see page 8, lines 5-13.

**P8L2 "We did not find any significant difference in total N% (2.9 _ 0.9 vs. 3.1 _ 0.7) or total C% (32.3 _ 2.8 vs. 34.8 _ 3.7)" suggest to move information also to M&M section**

We prefer to leave this sentence in this specific paragraph of the discussion because we make the point that although C and N content may not differ between cattle and pig slurry, there might be other biochemical differences between these slurries that ultimately are responsible for differences in C gains in soils.

**P2L31ff. ": : :As opposite in the control plots which do not receive any C and N addition, soil C accumulation may be partly explained by low C outputs via harvests and higher root mass production (possibly resulting from positive grass-legume interactions), which contributes to higher C inputs to soils when compared to the reduced root systems of fertilized slurry plots (see Fig. 5b).**

Plant biomass and thus plant C is removed in the same way across all plots. The amount of plant detritus returned to the soil every year is very small (plant litter mass is very low in all plots either control or fertilized plots). Control plots receive low C inputs via plant litter as the fertilized plots, thus we don't think that 'low C outputs' of control plots will lead to lower C inputs (through litter mass decomposition) than the fertilized plots. On page 9, lines 10-13 we state that: "In the control plots which do not receive any C addition, soil C accumulation may be partly explained by higher root mass (possibly resulting from positive grass-legume interactions), which contributes to higher C inputs to soils when compared to the reduced root systems of fertilized plots (see Fig. 5b)".

**P9L20 ": : :to beef management simply because dairy cattle are associated with higher CH4 emissions from enteric fermentation and manure management." Suppose that dairy cattle has also higher emissions due to the fact that for milking they stay more in the barn and less on the paddock as beef cows. IE barn emissions (manure, etc ) are higher for dairy cows**
This is a good point, in this case we used indications from IPCC guidelines but we agree with the reviewer that more detailed measurements should take into account differences in animal management within different farms.

---

## Author Comment (AC3) · 19 Jul 2016

**Schipper@waikato.ac.nz**

**Thanks for the responses.**

**In conclusions I dont agree that the previous work that I did suggests in the long-term pasture soils lose carbon. I think there needs to be a broader discussion here 'some' pasture systems will gain C and some will lose with time and many/most will likely be at steady state. Our previous work (Schipper et al 2014) demonstrated that the majority of soils in long-term pasture on flat land in New Zealand appear at steady state except for a couple of soil orders Gley (losses attributed to drainage and enhanced C respiration, as was suggested in Belgium) and Allophanic soils (we do not have an explanation for this loss).**

**In contrast pasture on hill country was gaining C, possibly due to recovery after past sheet wash erosion but not known. So in the end we need to understand the variety of pasture management systems that may increase or decrease soil C but a general statement that carbon increases or decreases in pastures is not really supported, it depends strongly on past land use and current management.**

We agree with the reviewer and we have modified this part in the conclusion section (see page 10, lines 14-18: "Previous studies suggests that the long-term net C balance of grassland soils will ultimately depend on how past and current management practices will influence losses and gains of C from/to soils (Schipper et al., 2014). It is likely that many grassland soils will eventually reach a C saturation point (Johnston et al., 2009; Smith 2014) where C losses and gains may still occur but could be relatively small compared to soil C stocks".

---

## Author Response (AR1)

**Response to Associate Editor**

Dear Prof. Michael Bahn,

We have now submitted a thoroughly revised manuscript where we included all editing suggestions by the two reviewers. In our previous responses to the reviewers we described how we followed their suggestions and we gave specific indications where we made the changes in our manuscript (page and line numbers). We thank the Editor and the reviewers' comments because they significantly contributed to improve our manuscript. We are confident that these major revisions have made our original research work of great interest to a large audience of scientists.

Best regards,

Dario